# Collagen polarization promotes epithelial elongation by stimulating locoregional cell proliferation

Hiroko Katsuno-Kambe[1], Jessica L Teo[1], Robert J Ju[1], James Hudson[2], Samantha J Stehbens[1], Alpha S Yap[1]*

[1]Division of Cell and Developmental Biology, Institute for Molecular Bioscience, The University of Queensland, Brisbane, Australia; [2]QIMR Berghofer Medical Research Institute, Brisbane, Australia

**Abstract** Epithelial networks are commonly generated by processes where multicellular aggregates elongate and branch. Here, we focus on understanding cellular mechanisms for elongation using an organotypic culture system as a model of mammary epithelial anlage. Isotropic cell aggregates broke symmetry and slowly elongated when transplanted into collagen 1 gels. The elongating regions of aggregates displayed enhanced cell proliferation that was necessary for elongation to occur. Strikingly, this locoregional increase in cell proliferation occurred where collagen 1 fibrils reorganized into bundles that were polarized with the elongating aggregates. Applying external stretch as a cell-independent way to reorganize the extracellular matrix, we found that collagen polarization stimulated regional cell proliferation to precipitate symmetry breaking and elongation. This required β1-integrin and ERK signaling. We propose that collagen polarization supports epithelial anlagen elongation by stimulating locoregional cell proliferation. This could provide a long-lasting structural memory of the initial axis that is generated when anlage break symmetry.

*For correspondence:
a.yap@uq.edu.au

Competing interest: The authors declare that no competing interests exist.

## Introduction

Branched tubules represent one of the archetypal modes of epithelial organization (*Iruela-Arispe and Beitel, 2013*; *Varner and Nelson, 2014*). In organs such as the mammary gland and lungs, networks of hollow epithelial tubes mediate the physiological exchange of gases, nutrients, and solutes between the body and its external environment (*Chung and Andrew, 2008*). These definitive networks are established by a complex morphogenetic process, where tubules grow outward from their precursors until they are instructed to branch, after which outgrowth continues until the network is completed (*Affolter et al., 2009*; *Andrew and Ewald, 2010*). The branching and elongation of multicellular aggregates can thus be considered as fundamental processes in the generation of tubular networks.

Interestingly, different organs also use distinct strategies for tubulogenesis. For example, in the trachea, lumens appear early and accompany the growth of tubules (*Schottenfeld et al., 2010*), whereas in the salivary and mammary glands tubules first begin as non-polarized cellular aggregates (or anlage), which then elongate as solid, multicellular cords before eventually forming lumens (*Bastidas-Ponce et al., 2017*; *Nerger and Nelson, 2019*; *Tucker, 2007*). In the present study, we use the mammary epithelium as a model to analyze what guides the elongation of multicellular anlage.

Tubulogenesis is a highly regulated phenomenon. The decision to branch is recognized to be a critical checkpoint that is controlled by developmental signals and cell-cell and cell-extracellular matrix (cell-ECM) interactions (*Goodwin and Nelson, 2020*). The elongation of tubule precursors is also thought to be a regulated process controlled by receptor tyrosine kinases and other signaling pathways (*Costantini and Kopan, 2010*; *Gjorevski and Nelson, 2010*; *Sternlicht et al., 2006*). However,

elongation occurs over hours to days, time scales that are much longer than the underlying cellular processes and the signaling pathways that guide them. This raises the question of whether there may be mechanisms that can help guide elongation over longer time scales, effectively serving as a bridge between the rapidity of cell signaling and the slow progression of macroscopic anlage elongation. In this study, we show how collagen 1 within the ECM can provide such a bridge by stimulating cell proliferation.

The ECM comprises complex mixtures of proteins, glycosaminoglycans, and glycoconjugates that fill the extracellular spaces of tissues and organs (*Frantz et al., 2010*). One of the main components of ECM is collagen, particularly type 1 collagen (also called collagen 1), which is often the dominant form during epithelial tubulogenesis (*Graham et al., 1988*; *Keely et al., 1995*; *Llacua et al., 2018*; *Nakanishi et al., 1986*; *Simon-Assmann et al., 1995*). Type 1 collagen exerts a diverse range of effects that can potentially influence epithelial elongation. Fibrillar collagen helps scaffold other molecules, such other ECM proteins and growth factors (*Kanematsu et al., 2004*; *Wipff and Hinz, 2008*). Adhesion between cells and ECM allows the chemical and mechanical properties of the ECM to regulate cell signaling and gene regulation (*Shi et al., 2011*). These cell-ECM adhesions also allow cell-based forces to reorganize the ECM. In particular, cells can rearrange collagen fibrils to influence cell migration (*Buchmann et al., 2021*; *Gjorevski et al., 2015*; *Guo et al., 2012*; *Shi et al., 2014*).

Importantly, fibrillar components of the ECM, such as collagen 1, are relatively long-lived (*Price and Spiro, 1977*; *Verzijl et al., 2000*), making them attractive candidates to bridge time scales during the elongation process. Indeed, rearrangement of collagen has been implicated in patterning epithelial branching (*Brownfield et al., 2013*; *Guo et al., 2012*; *Harunaga et al., 2011*; *Ingman et al., 2006*; *Patel et al., 2006*). However, it is difficult to elucidate the specific contribution for collagen in the complex environment of an organ, where there are many additional contributions from other cell types and various chemotactic signals. Therefore, in this study we used three-dimensional (3D) organotypic cultures to test how the ECM regulates epithelial elongation. We report that a collagen 1 matrix induces the elongation of mammary epithelial anlage. This is accompanied by the polarization of the matrix itself, an event that sustains anlagen elongation by stimulating cell proliferation.

## Results

### An experimental system to capture symmetry breaking and elongation of epithelial aggregates

To model the elongation of epithelial anlage, we induced multicellular aggregates to break symmetry by manipulating the ECM environment in which the cells were grown. When MCF10A cells are grown on Matrigel substrates and in media supplemented with soluble Matrigel (*Debnath et al., 2003*), they proliferate from single isolated cells to form a multicellular aggregate, which then polarizes and clears the central mass of apoptotic cells, generating a central lumen in a process known as cavitation (*Figure 1—figure supplement 1A*). In contrast, when isolated cells were embedded in a gel of type 1 collagen, which is the major ECM component in the stromal environment during mammary tubulogenesis (*Schedin and Keely, 2011*), they proliferated to form elongated, solid cords of non-polarized cells (*Figure 1—figure supplement 1B*; *Krause et al., 2008*). This indicated that some properties of the collagen 1 environment might provide an instructive cue for elongation.

In order to capture the process by which isotropic aggregates broke symmetry and then elongated, cells were first seeded in Matrigel until proliferation arrested at 10 days (*Figure 1A*); aggregates were then isolated and embedded into collagen 1 (acid-solubilized rat type I collagen, 37°C for 30 min). Lumens were apparent in some Matrigel-embedded aggregates, but most remained as solid spheres, resembling anlage. Live-cell imaging revealed that the transplanted aggregates displayed small jiggling motions for several hours, then spontaneously broke symmetry and elongated to form cord-like structures similar to those formed when cultured in collagen from the outset (*Figure 1B*, *Video 1*). These elongating aggregates were generally solid, with no evident lumens (*Figure 2A*). Commonly, elongation began with a group of cells that protruded away from the more spherical original aggregate (*Figure 1C and E*). In contrast, when aggregates were transplanted from Matrigel back into Matrigel, they grew slightly but did not elongate (*Figure 1—figure supplement 1C*). Therefore, transplantation into a collagen 1 matrix effectively caused isotropic MCF10A aggregates to break symmetry and elongate.

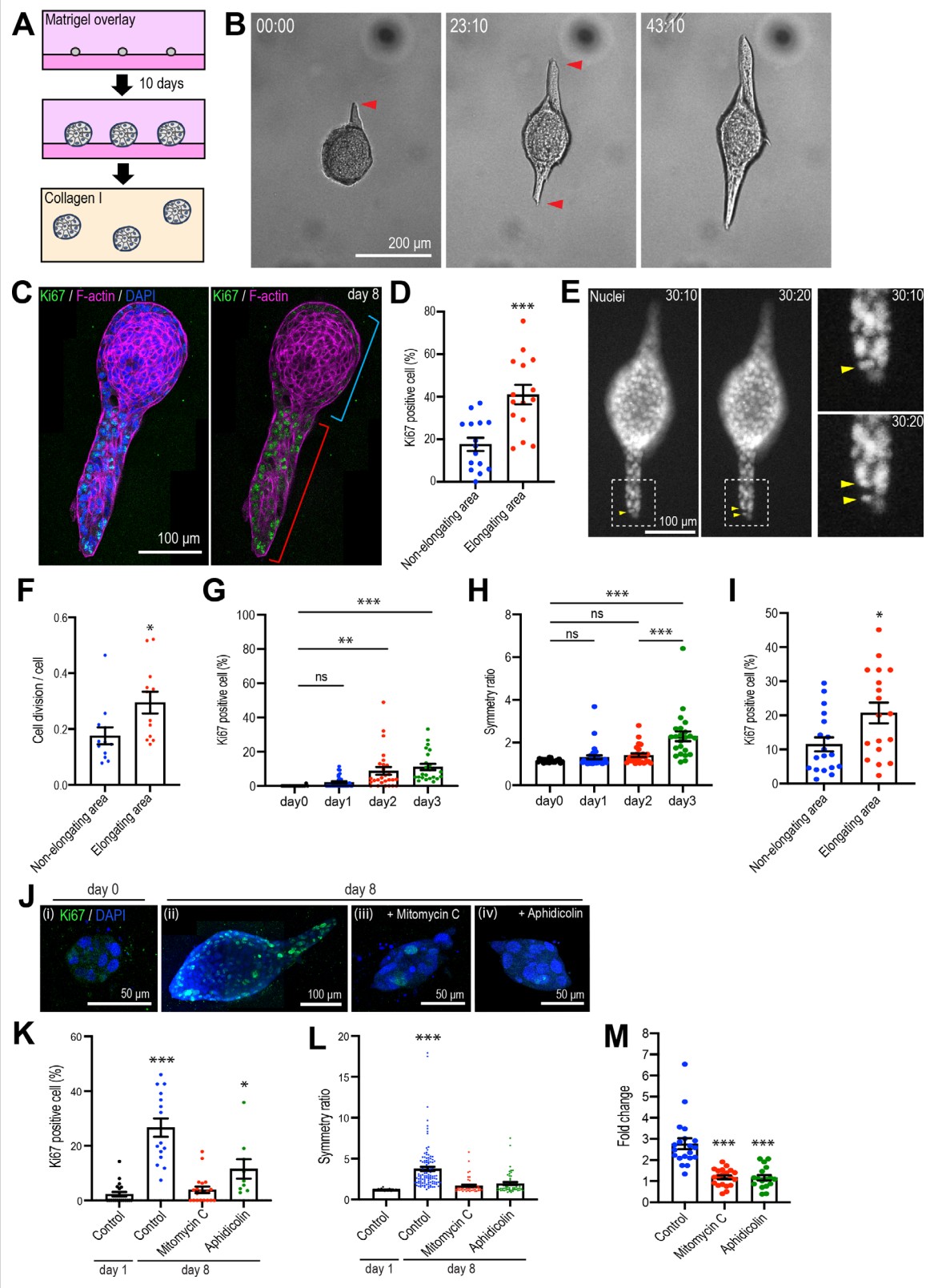

**Figure 1.** Type 1 collagen induces elongation of MCF10A anlage via cell proliferation. (**A**) Cartoon: transplantation of MCF10A cell aggregates from Matrigel into collagen gel. Single isolated cells are cultured on Matrigel and overlaid with Matrigel-containing medium. Matrigel is washed out after 10 days and the aggregates re-embedded into type 1 collagen gel. (**B**) Time-lapse images of MCF10A aggregates after transfer into a collagen gel. Red arrowheads: elongation from aggregates. (**C**) Fluorescence image of an elongated aggregate cultured for 8 days after transfer into collagen gel and

*Figure 1 continued on next page*

*Figure 1 continued*

stained with anti-Ki67 antibody (green), phalloidin (magenta), and DAPI (blue). Blue parenthesis: non-elongating area; red parenthesis: elongating area. (**D**) Percentage of Ki67-positive cells in elongating and non-elongating areas of the elongated aggregates (n = 15 aggregates). (**E**) Time-lapse images of elongating aggregates expressing NLS-mCherry. Yellow arrowheads: dividing nucleus. (**F**) Frequency of cell division in elongating and non-elongating regions of aggregates (n = 12 aggregates). (**G**) Percentage of Ki67-positive cells in aggregates cultured for 0–3 days (n = 102 aggregates). (**H**) Symmetry ratio of aggregates cultured for 0–3 days (n = 102 aggregates). (**I**) Percentage of Ki67-positive cells in elongating area and non-elongating areas of aggregates that broke symmetry (defined as symmetry ratio >1.5) in the first three days of culture (n = 18 aggregates). (**J**) MCF10A aggregates co-stained with anti-Ki67 antibody (green) and DAPI (blue). Aggregates were cultured for 0 day (**i**) or 8 days (**ii–iv**) after treatment with vehicle (**ii**), mitomycin C (**iii**), or aphidicolin (**iv**). (**K**) Percentage of Ki67-positive cells in aggregates cultured for 1 day or 8 days with or without mitomycin C or aphidicolin (n = 63 aggregates). (**L**) Symmetry ratio of aggregates cultured for 1 day or 8 days with or without mitomycin C or aphidicolin (n = 237 aggregates). (**M**) Effect of delayed inhibition of proliferation on aggregate elongation. Aggregates were cultured for 3 days before treatment with mitomycin C or aphidicolin. Data are fold change of elongation in control and drug-treated cultures (n = 57 aggregates). All data are means ± SEM, *p<0.05, **p<0.01, ***p<0.001. Data in (**D, F, I**) were analyzed by unpaired Student's *t*-test. Data in (**G, H, K–M**) were analyzed by one-way ANOVA Tukey's multiple comparisons test.

The online version of this article includes the following source data and figure supplement(s) for figure 1:

**Source data 1.** Original data for quantitative analysis in *Figure 1*.

**Figure supplement 1.** 3D morphology of MCF10A cells.

**Figure supplement 2.** Cell proliferation and division during aggregate elongation.

**Figure supplement 3.** Cell motility during aggregate elongation.

## Cell proliferation drives anlagen elongation

Multiple cellular processes have been implicated in epithelial elongation (*Andrew and Ewald, 2010*; *Economou et al., 2013*; *Keller, 2002*). We first examined cell proliferation, given the evident increase in the size of aggregates that elongated. Staining for Ki67, a nonhistone nuclear protein commonly used as a marker of proliferating cells (*Soliman and Yussif, 2016*), showed clear proliferation in aggregates that had elongated 8 days after transplantation into collagen (*Figure 1C and D*). These elongating aggregates commonly consisted of elongated extensions as well as a rounded region that marked the original aggregate (*Figure 1B*). Strikingly, Ki67-positive cells were more frequent in the elongating parts of the aggregates (41.00% ± 4.55% of all cells) compared with the non-elongating parts of the aggregates (18.86% ± 3.05%) (*Figure 1C and D*). This differential pattern of proliferation was also evident when we analyzed just the cells at the surface of the aggregates, which were in contact with the collagen (*Figure 1—figure supplement 2A*). Live-cell imaging of cells expressing NLS-mCherry also showed that the proportion of nuclei that divided was higher in the elongating than the non-elongating areas within aggregates (*Figure 1E and F*). Thus, elongation was associated with locoregional differences in cell proliferation, which was enhanced in those parts of the aggregates that elongated.

More detailed inspection revealed that increased proliferation was first evident in aggregates 2 days after transplantation (*Figure 1G*) and appeared to precede the onset of symmetry breaking (*Figure 1H*). To quantitate aggregate elongation, we calculated a symmetry ratio, that is, the ratio of the maximum length ($L_1$) and width ($L_2$) of the aggregates. Completely round aggregates will have a symmetry ratio of 1, whereas for elongated aggregates $L_1/L_2$ will be >1 (*Figure 2B*). The symmetry ratio first increased at day 3 (*Figure 1H*, *Figure 1—figure supplement 2C*), approximately 24 hr after an increase in proliferation was detected (*Figure 1G*). To test

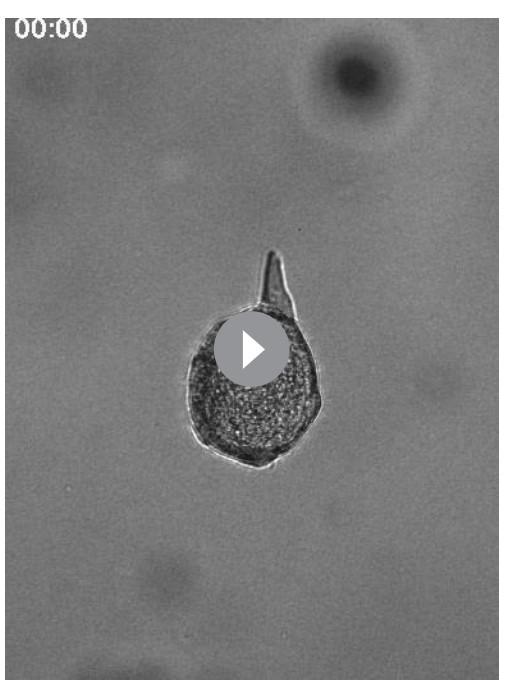

**Video 1.** Time-lapse images of MCF10A aggregates after transferred into collagen gel. Images were taken every 10 min.

https://elifesciences.org/articles/67915/figures#video1

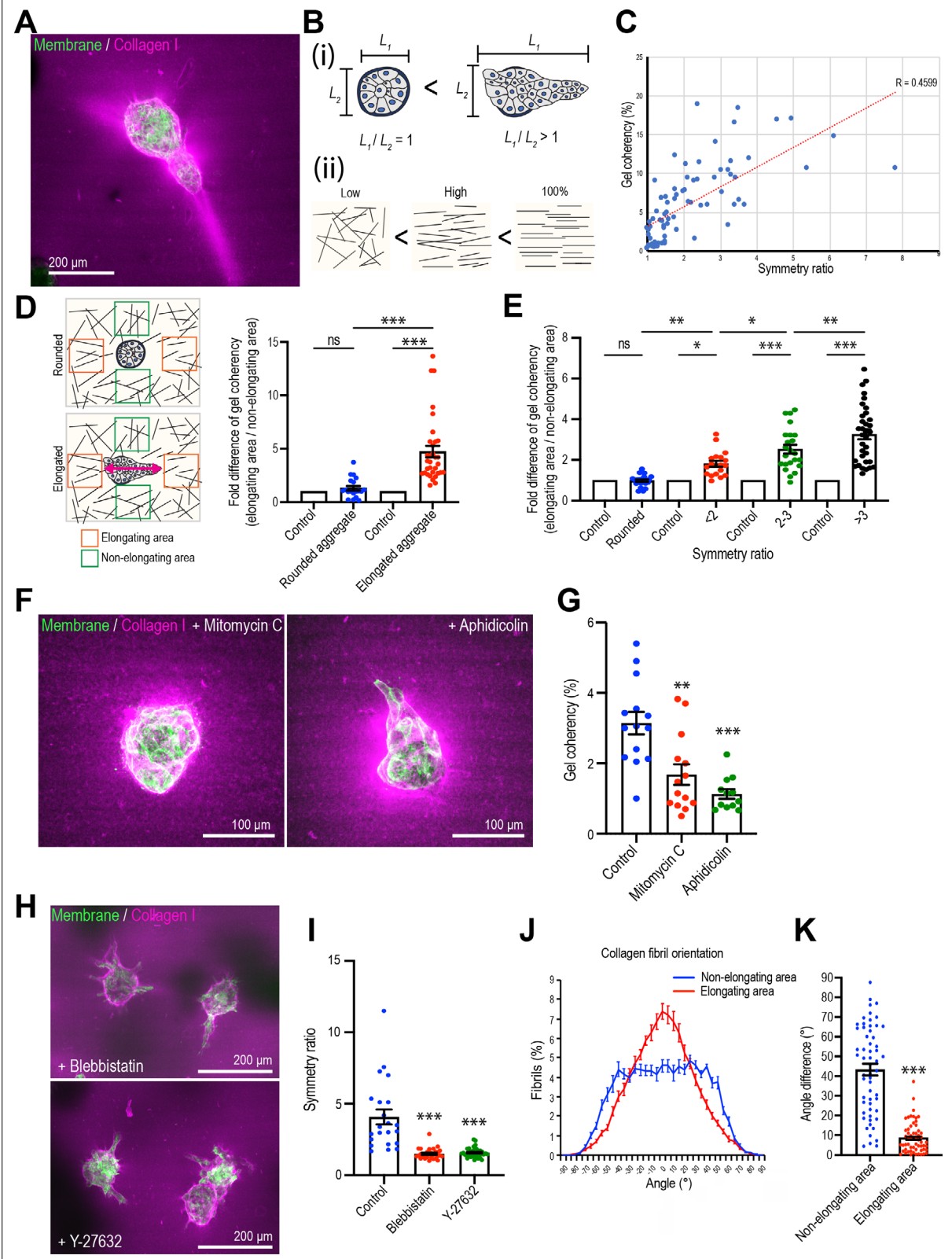

**Figure 2.** Mammary cell aggregates polarize the extracellular matrix (ECM) as they elongate. (**A**) Fluorescent image of collagen fiber alignment and elongated aggregates expressing the cell membrane marker GFP-HRasC20. Collagen fibers are labeled with mCherry-CNA35 peptide (magenta). (**B**) Cartoon of symmetry ratio of aggregates (i) and gel coherency (ii). (**C**) Scatter plot of aggregate symmetry ratio and collagen fiber coherency (n = 75 aggregates). (**D**) Regional analysis of collagen coherency around aggregates. Cartoon illustrates the approach: for elongating aggregates, coherency

*Figure 2 continued on next page*

*Figure 2 continued*

was measured in regions of interest (ROIs) placed both at the tips of elongations and proximate to their non-rounded areas. For rounded aggregates, ROIs were placed orthogonally. Regional differences in coherency were measured as the fold difference, measured around rounded aggregates (n = 19 aggregates) and elongated aggregates (n = 57 aggregates). Elongated aggregates were defined as symmetry ratio >1.5. Double-headed arrow: elongating axis. (**E**) Fold difference of collagen fiber coherency around aggregates measured at early stages of elongation (first three days of culture) subdivided based on symmetry ratio (n = 98 aggregates). (**F**) Fluorescent images of collagen fibers labeled with mCherry-CNA35 (magenta) and aggregates expressing GFP-HRasC20 cultured for 8 days with mitomycin C or aphidicolin. (**G**) Coherency of collagen fibers surrounding the aggregates treated with mitomycin C and aphidicolin (n = 40 aggregates). (**H**) Fluorescent images of collagen fiber alignment with aggregates expressing GFP-HRasC20 treated with blebbistatin or Y-27632 for 8 days. Collagen fibrils were labeled with CNA35-mCherry (magenta). (**I**) Symmetry ratio of aggregates cultured for 8 days with or without blebbistatin or Y-27632 (n = 88 aggregates). (**J**) Distribution of collagen fiber orientation surrounding elongated aggregates (n = 22 aggregates) or non-elongated aggregates (n = 13 aggregates). (**K**) Difference between elongation axis of aggregates and average angle of collagen fibers in non-elongating area or elongating area (n = 57 aggregates). All data are means ± SEM, ns, not significant, *p<0.05, **p<0.01, ***p<0.001. Data in (**D, E, G, I**) were analyzed by one-way ANOVA Tukey's multiple comparisons test. Data in (**K**) were analyzed by unpaired Student's *t*-test.

The online version of this article includes the following source data and figure supplement(s) for figure 2:

**Source data 1.** Original data for quantitative analysis in *Figure 2*.

**Figure supplement 1.** Immunofluorescent staining of extracellular matrix (ECM) proteins in the aggregates.

**Figure supplement 2.** Rac1 GEF inhibitor does not affect MCF10A aggregate elongation.

if any locoregional differences in proliferation were to be found at these early stages of symmetry breaking, we analyzed aggregates that had broken symmetry (defined experimentally as a symmetry ratio >1.5) after 3 days of culture. We compared the proportion of cells that were Ki67-positive in the elongating regions of aggregates with the proportion to be found in the non-elongating areas of the same aggregates (*Figure 1I*, *Figure 1—figure supplement 2D*). Ki67 positivity was twofold greater in the elongating areas (*Figure 1I*), suggesting that locoregional differences in proliferation were established early in the elongation process.

Importantly, elongation was inhibited when proliferation was blocked using the cell cycle inhibitors mitomycin C and aphidicolin (*Figure 1J*). In control cultures, the proportion of Ki67-positive cells increased progressively after transplantation into collagen (from 2.28% ± 0.85% at day 1 to 26.63% ± 3.34% at day 8). However, cell proliferation was significantly reduced by mitomycin C (3.88% ± 1.18%, day 8) and aphidicolin (11.54% ± 3.55%, day 8) (*Figure 1K*). Aggregate elongation was also significantly reduced by both mitotic inhibitors (*Figure 1J*). Whereas the symmetry ratio increased as control anlage elongated, this was blocked by both mitomycin C and aphidicolin (*Figure 1L*). Formally, cell proliferation might have been necessary to initiate symmetry breaking and cell elongation or ongoing proliferation might also have served to sustain elongation. To pursue this, we added mitomycin C or aphidicolin after the first signs of symmetry breaking were evident (at 3 days after transplantation). This reduced elongation by approximately threefold (*Figure 1M*). Therefore, cell proliferation was required for elongation, with the capacity to contribute early in the symmetry-breaking process and also later to sustain elongation.

We then used live-cell imaging of labeled nuclei to evaluate other processes implicated in epithelial elongation. Cells within the elongating regions tended to divide along the axis of elongation (*Figure 1—figure supplement 2B*), as revealed by comparing the axis of cell division and the principal axis of the region. The angle difference between these axes was smaller in elongating areas compared with non-elongating areas within the same aggregate (*Figure 1—figure supplement 2B*). This suggested that polarized cell division accompanied enhanced proliferation during aggregate elongation (*Gong et al., 2004*; *Keller, 2006*).

Tracking also revealed that cells were motile within elongating aggregates (*Figure 1—figure supplement 3A*). This was evident in rounded aggregates that had not broken symmetry as well as in aggregates that had elongated (*Figure 1—figure supplement 3B*). When we examined regional differences within aggregates that had undergone elongation, we found that the speeds of migration were identical in the parts that were elongating compared with the non-elongating regions of the aggregates (*Figure 1—figure supplement 3C*). However, the straightness of the tracks, which was used as an index of the persistence of migration, was slightly greater in the elongating areas than in the non-elongating areas (*Figure 1—figure supplement 3D*). Furthermore, cells within regions of elongation appeared to orient better with the axis of the aggregate than did cells found in non-elongating

areas. We measured this by comparing the orientation of the tracks with the principal axis of the aggregates (track displacement angle, *Figure 1—figure supplement 3E*). The track displacement angle was less in the elongating areas than in the non-elongating areas.

Inhibition of proliferation with either mitomycin C or aphidicolin did not affect cell migration speeds in either the rounded or elongated areas (*Figure 1—figure supplement 3B*). However, blocking proliferation slightly reduced track straightness, implying a reduction in persistence (*Figure 1—figure supplement 3F*) and increased the track displacement angle (*Figure 1—figure supplement 3G*). Therefore, the apparently orderly migration of cells was compromised by blocking cell proliferation. Together, these results suggest that proliferation was essential for the elongation process and may also have influenced aspects of directional migration that would be predicted to reinforce the elongation process.

## Collagen 1 condenses around epithelial aggregates that break symmetry

To gain further insight into the process of elongation, we sought to identify changes in the ECM during this process. Consistent with the change in matrix environment associated with the transplantation process, re-embedded aggregates lost laminin V expression over time (*Figure 2—figure supplement 1A*). They expressed fibronectin throughout the experiments without any evident regional differences in the elongating areas of aggregates (*Figure 2—figure supplement 1B*).

Then we visualized collagen 1 by labeling with the collagen-binding peptide CNA35 (mCherry-CNA35; *Krahn et al., 2006*), mixed with soluble collagen 1 before the incorporation of MCF10A aggregates (*Figure 2A*). Collagen 1, in contrast to the basement membrane-like Matrigel, is composed of fibrillar polymers that can guide cell movement and whose organization is influenced by the application of cellular forces (*Brownfield et al., 2013*; *Gjorevski et al., 2015*; *Piotrowski-Daspit et al., 2017*). Aggregate elongation appeared to coincide with change in fibril organization. Whereas fibrils appeared isotropic in acellular gels, collagen condensed around the aggregates that had begun to elongate, forming dense bands that extended away from the cells into the gel. However, it was seldom possible to visualize individual fibrils with the low-magnification, long-working distance lenses that were required to visualize aggregates within the gels.

Therefore, we measured the coherency (or co-orientation) of gels with Orientation J, which calculates the local orientation and isotropy for each pixel in an image based on the structure tensor for that pixel (see Materials and methods for details; *Rezakhaniha et al., 2012*). We interpret coherency as reflecting collagen bundling and condensation (which we shall call 'bundling' for short), such as has been observed elsewhere during elongation (*Brownfield et al., 2013*; *Buchmann et al., 2021*; *Gjorevski et al., 2015*). Then, we compared collagen coherency with aggregate shape, as measured by the symmetry ratio (*Figure 2B*), after 8 days culture, when elongation was established. Overall, collagen coherency increased with the increase in symmetry ratio (*Figure 2C*), implying that the degree of collagen bundling increased with aggregate elongation. Furthermore, closer examination around elongating aggregates showed that collagen coherency was greater in regions proximate to the elongating parts of the aggregates compared with the non-elongating parts (*Figure 2D*). In contrast, aggregates that remained spherical showed no regional differences in collagen organization. Similarly, earlier studies showed that collagen reorganized ahead of the tips of elongating aggregates (*Brownfield et al., 2013*; *Gjorevski et al., 2015*). This suggested that collagen became increasingly bundled and condensed where aggregates were elongating.

Increased collagen bundling was also evident at the early stages of elongation. Because aggregates varied in the timing of when they broke symmetry, we examined aggregates by live imaging in the first three days of the assays. We then subdivided these based on their symmetry ratio and compared the coherency of collagen 1 around the elongating areas of aggregates with that around the non-elongating areas (i.e., those that remained rounded, expressed as a fold difference). This showed that collagen 1 coherency increased around the elongating areas as they broke symmetry (*Figure 2E*). Thus, from the early stages of symmetry breaking collagen 1 bundling increased around the sites where aggregates elongated.

Collagen bundling required cell proliferation, as coherency was reduced by treatment with either mitomycin C or aphidicolin (*Figure 2F and G*). Collagen bundling was also reduced when we inhibited cellular contractility with the myosin antagonist, blebbistatin, or the Rho kinase (ROCK) inhibitor,

Y27632 (*Figure 2H, I*), consistent with reports that cell contractility can condense collagen (*Brownfield et al., 2013*; *Buchmann et al., 2021*; *Gjorevski et al., 2015*). In contrast, collagen condensation persisted around aggregates treated with the Rac1 GEF inhibitor NSC23766, which did not affect aggregate elongation or the speed with which cells moved within the aggregates (*Figure 2—figure supplement 2*). This suggested that aggregates exerted forces to reorganize their local collagen environment as they broke symmetry.

## Collagen polarity co-aligns with cell aggregates during elongation

We then asked if higher order organization in the collagen 1 gels was altered as aggregates elongated. For this, we used Orientation J to extract the principal axis of orientation in the gel, as a measure of its polarity, around elongating or non-elongating parts of an aggregate. Then we compared these gel orientations with the axis of elongation of the cell aggregate. First, we assembled histograms of gel orientation, setting the principal axis of elongation in the whole aggregate as 0° (*Figure 2J*). The gels around non-elongating areas of the aggregates showed a broad distribution of orientations relative to the axis of the aggregate (*Figure 2J*). In contrast, the gel around the elongating regions of aggregates tended to orient with the principal axis of the aggregates. This suggested that the gel preferentially co-aligned with the aggregates around the regions of elongation. This notion was reinforced by comparing the angle differences between the principal axis (polarity) of the gel and of the aggregates. In this analysis, a decrease in the difference in angles between these two axes indicates an increase in their co-alignment. We found that the angle differences between the axes of collagen polarity and aggregate elongation was significantly smaller around the elongating regions than around the non-elongating regions of the aggregates (*Figure 2K*). Thus, the collagen matrix became polarized and co-aligned with the aggregates around regions of elongation.

## Collagen polarization stimulates cell proliferation to induce aggregate elongation

This led us to wonder if polarization of the collagen could itself influence the process of aggregate elongation. To test this, we developed a strategy that applied exogenous stretch to polarize the matrix independently of cell-generated forces. In this protocol, a collagen gel was created within a ring-shaped polydimethylsiloxane (PDMS) frame, then stretched uniaxially by expanding a stretcher inserted into the hole of the ring for 4 hr (*Figure 3—figure supplement 1A and B*). Second harmonic imaging confirmed that collagen fibers were isotropically distributed in the unstretched gels and became more coherent (bundled) and more polarized immediately after stretching (*Figure 3—figure supplement 1C and D*), with their principal axes oriented in the direction of stretch (*Figure 3—figure supplement 1E*). Moreover, polarization of the collagen fibrils could be preserved for at least 7 days after the application of stretch by re-embedding the collagen rings in a larger collagen gel (*Figure 3A–C*, *Figure 3—figure supplement 1B*). In contrast, when gels were allowed to float in medium after stretching (*Figure 3D*, *Figure 3—figure supplement 1B*), gel coherency (*Figure 3E*) and polarity (*Figure 3F*) reverted to that of unstretched gels. Thus, re-embedding allowed us to sustain collagen polarization for a prolonged period after the initial stretching.

We then transplanted spherical cell aggregates from Matrigel into the collagen rings and applied our stretching protocol. Interestingly, the aggregates did not become elongated during the period when stretch was actively applied. The symmetry ratios of aggregates immediately after stretching were identical to those in unstretched aggregates (*Figure 3—figure supplement 2A and B*). Instead, aggregates broke symmetry many hours after the period of active stretching. However, symmetry breaking occurred earlier when stretched gels were allowed to retain collagen polarization by re-embedding (35.14 ± 3.28 hr, *Figure 3G and H*) compared with aggregates that were allowed to break symmetry spontaneously in non-stretched gels (68.40 ± 5.88 hr, p<0.001). Moreover, the degree of elongation was also enhanced once it had begun. The symmetry ratios of aggregates embedded in stretch-polarized gels were significantly greater than those seen with native gels (*Figure 3I*). Therefore, polarization of the collagen gel could stimulate the cell aggregates to break symmetry.

Stretch-polarized aggregates also elongated preferentially along the axis of the exogenous stretch, whereas aggregates in non-stretched gels elongated in a random direction (*Figure 3G and*

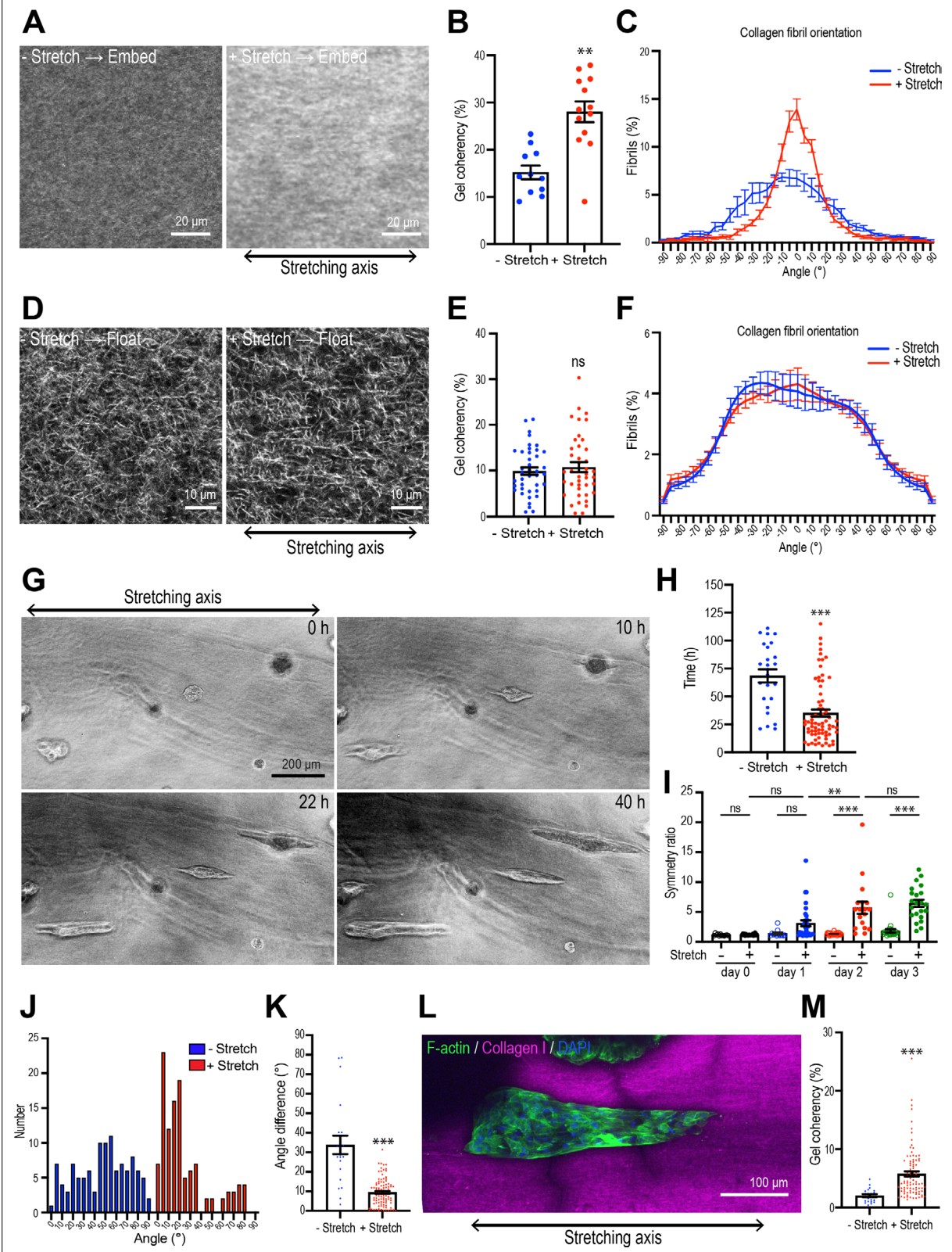

**Figure 3.** Collagen polarization induces mammary aggregate elongation. (**A**) Second harmonic generation (SHG) images of collagen fibers in the gel with or without stretching and incubated for 7 days after re-embedding in gel. Double-headed arrow: stretching axis. (**B**) Coherency of collagen fiber in the gel with or without stretching (n = 24 positions in multiple gels). (**C**) Distribution of collagen fiber orientation in the gel with or without stretching. 0° is defined as the axis of stretch (N = 3 independent experiments). (**D**) SHG images of collagen fiber floated for 7 days with or without after stretching.

*Figure 3 continued on next page*

*Figure 3 continued*

Double-headed arrow: stretching axis. (**E**) Coherency of collagen fiber in 7 days floated gel with or without stretching (n = 46 positions in multiple gels). (**F**) Distribution of collagen fiber orientation in gels that had been allowed to float (7 days) with or without prior stretching (N = 3 independent experiments). (**G**) Time-lapse images of aggregates embedded in stretched gel. Double-headed arrow: stretching axis. (**H**) Initiation time of aggregate elongation in the gel with or without stretching (– stretch: n = 25 aggregates, +stretch: n = 71 aggregates). (**I**) Symmetry ratio of aggregates in the early phase of culture (0–3 days) with or without stretching (–stretch, day 0: n = 12; day 1: n = 12; day 2: n = 24; day 3: n = 24; +stretch, day 0: n = 21; day 1: n = 32; day 2: n = 17; day 3: n = 22). (**J**) Distribution of elongation axes of aggregates in the gel with or without stretching (–stretch: n = 112 aggregates, +stretch: n = 115 aggregates). (**K**) Difference between elongating axis of aggregates and average angle of collagen fibers in the gel with or without stretching (–stretch: n = 21 aggregates, +stretch: n = 102 aggregates). (**L**) Fluorescence image of aggregates cultured for 5 days after gel stretching and co-stained with phalloidin (green) and DAPI (blue) in stretched gel. Collagen fibrils were labeled with CNA35-mCherry (magenta). (**M**) Coherency of collagen fibers surrounding elongated aggregates in the gel with or without stretching (–stretch: n = 21 aggregates, +stretch: n = 102 aggregates). All data are means ± SEM; ns, not significant, **p<0.01, ***p<0.001. Data in (**B, E, H, K, M**) were analyzed by unpaired Student's *t*-test. Data in (**I**) were analyzed by one-way ANOVA Tukey's multiple comparisons test.

The online version of this article includes the following source data and figure supplement(s) for figure 3:

**Source data 1.** Original data for quantitative analysis in *Figure 3*.

**Figure supplement 1.** External gel stretching aligns collagen fiber.

**Figure supplement 2.** MCF10A aggregates elongate along the gel stretching axis.

*J*, *Figure 3—figure supplement 2C and D*, *Video 2*). The angle difference between collagen fiber polarity and aggregate elongation was significantly smaller in stretch-polarized gels than in unstretched control gels (*Figure 3K*). Of note, collagen gel reorganization was preserved in the stretch-polarized gels (*Figure 3L and M*), even in the presence of cellular aggregates, and this was oriented in the direction of the original stretch. This suggested that in this assay the elongating aggregates were following the polarity of the gel itself. In contrast, similar intensity of fibronectin staining was seen in stretch-polarized aggregates compared with controls, suggesting that fibronectin deposition was not stimulated by the stretch stimulation (*Figure 3—figure supplement 2E*).

Importantly, the impact of stretch on the aggregates was disrupted when collagen polarization was reversed. The proportion of aggregates that elongated after stretching was reduced when the gels were floated rather than being re-embedded (*Figure 4A*) and their length of elongation was reduced (*Figure 4B*). Of note, cell-based forces could not overcome the external forces acting on the gel as the reversal of collagen polarization occurred even when aggregates were incorporated into the gel (*Figure 4C*). This implied that collagen polarization may have been responsible for allowing external stretch to promote aggregate elongation.

To confirm that these results were due to a critical role of collagen polarization, rather than as-yet-unknown impacts of stretch upon the cells, we used collagenase to extract aggregates from gels immediately after their 4 hr stretch, then transplanted them into unstretched gels where the collagen fibers oriented randomly (*Figure 4D*). We reasoned that if elongation were due to a cell-intrinsic mechanism that bore the memory of the stretch, then this should be preserved even after cells were transplanted into a naïve gel. As noted earlier, aggregate elongation occurred earlier when stretched gels were re-embedded. However, this effect was lost when cells were removed from the stretched gel immediately after stretching and transplanted into a naïve isotropic gel (*Figure 4E*). Together, these results indicate that collagen polarization can direct aggregate elongation to accelerate its initiation and orient the direction of symmetry breaking.

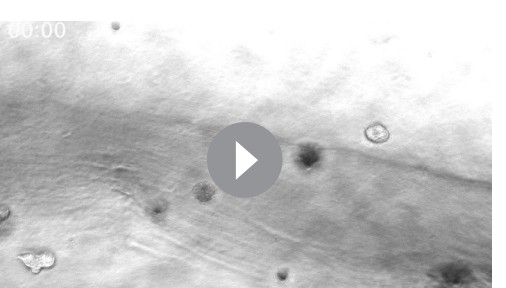

**Video 2.** Time-lapse images of aggregates embedded in stretched gel. Images were taken every 1 hr.
https://elifesciences.org/articles/67915/figures#video2

## Collagen matrix polarization stimulates cell proliferation for anlagen elongation

Since cell proliferation was a major driver of spontaneous aggregate elongation, we then asked if it was altered when the collagen gel was polarized by stretch. Indeed, we found that stretch polarization stimulated cell proliferation (*Figure 5A*, *Figure 5—figure supplement 1*). An increase in

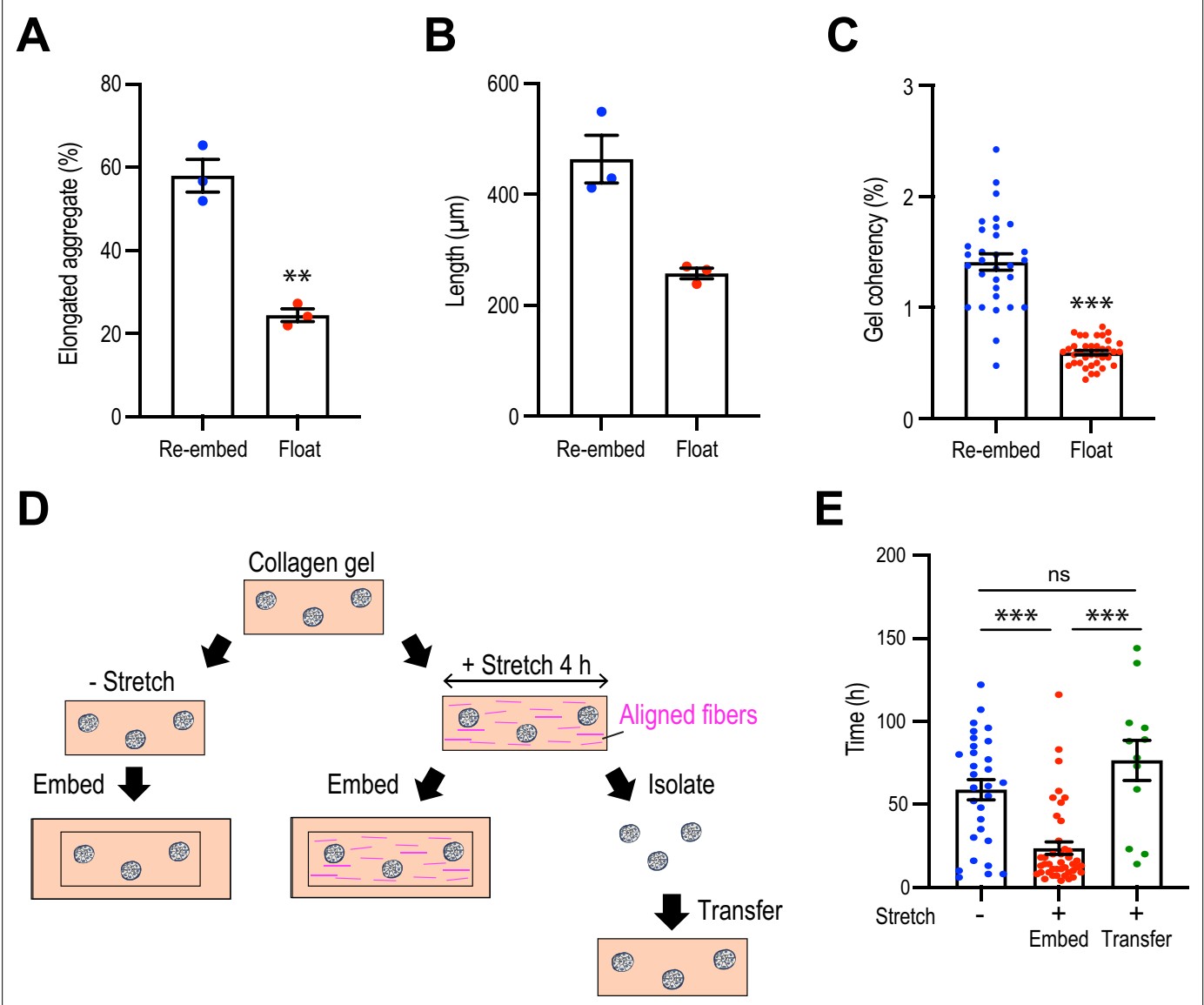

**Figure 4.** Collagen polarization must be sustained to stimulate mammary aggregate elongation. (**A**) Population of elongated aggregates in the re-embedded or floated gel (N = 3 independent experiments). (**B**) The length of elongated aggregates in the re-embedded or floated gel (N = 3 independent experiments). (**C**) Coherency of collagen fibers surrounding the aggregates in gels that were re-embedded after stretch or floated for 7 days after stretching (n = 68 aggregates). (**D**) Cartoon of aggregate from stretched gel into normal collagen gel. Aggregates were isolated from stretched gel by collagenase and re-embedded in naïve gel. (**E**) Initiation time for aggregate elongation in non-stretched control gels, stretched gels that had been re-embedded to preserve collagen polarization (embed) and after cells were extracted and transferred into non-stretched gels (transfer) (n = 83 aggregates). All data are means ± SEM; ns, not significant, **p<0.01, ***p<0.001. Data in (**A–C**) were analyzed by unpaired Student's *t*-test. Data in (**E**) were analyzed by one-way ANOVA Tukey's multiple comparisons test.

The online version of this article includes the following source data for figure 4:

**Source data 1.** Original data for quantitative analysis in *Figure 4*.

the proportion of Ki67-positive cells was evident at day 1 after stretch polarization, whereas it did not increase till day 2 in unstretched gels. Moreover, the proportion of Ki67 cells was consistently greater in stretch-polarized gels during the early phase of culture (days 1–3) as well as at the end of our experiments (day 8). Cell proliferation mediated the enhanced elongation in the stretched gels as both the length (*Figure 5B*) and symmetry ratio (*Figure 5C*) of stretch-stimulated aggregates were reduced by mitomycin C and aphidicolin.

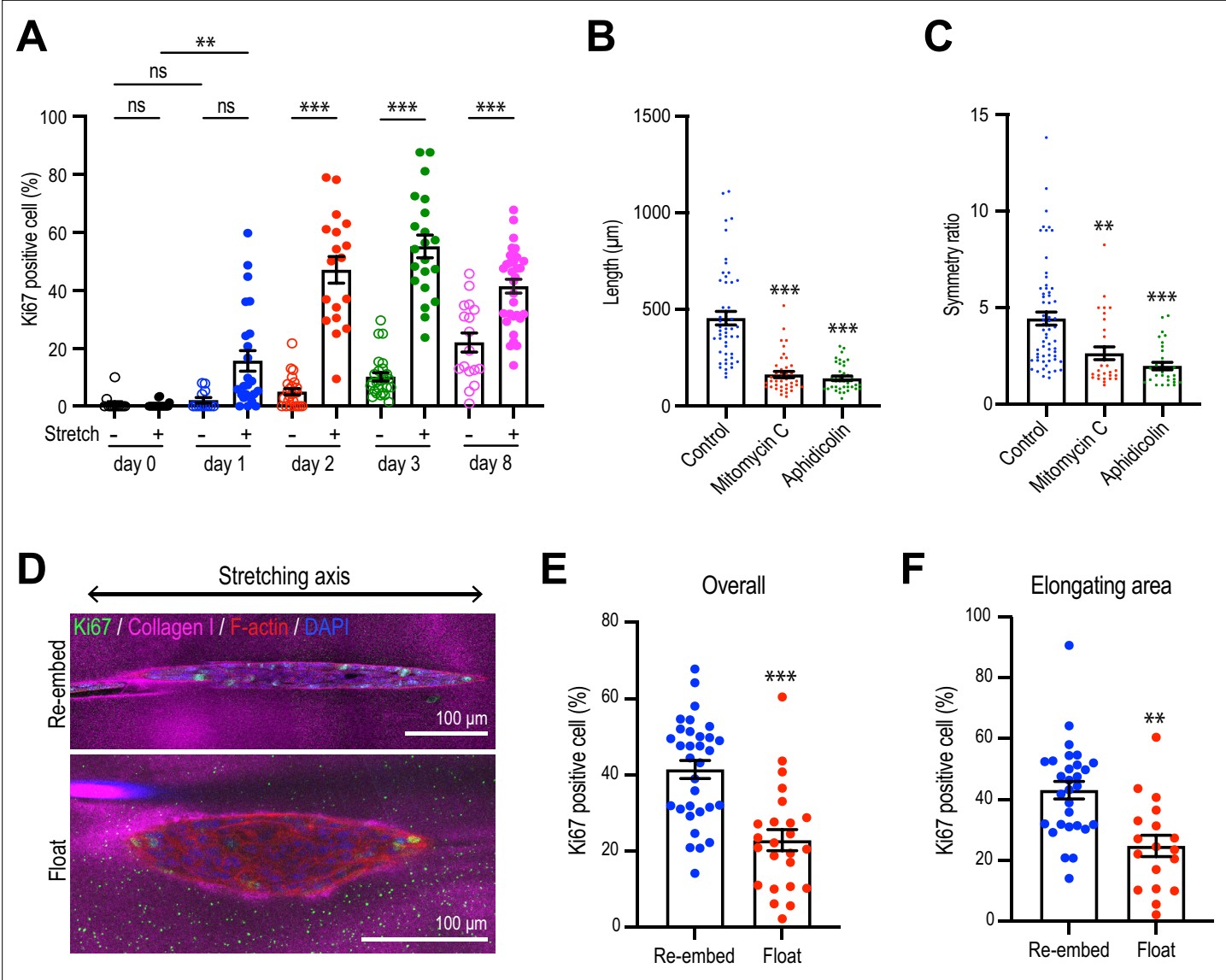

**Figure 5.** Collagen polarization induces cell proliferation for aggregate elongation. (**A**) Time course of cell proliferation within aggregates in control gels (–stretch) or after stretching (+stretch). Data were percentage of cells that were Ki67-positive (n = 206 aggregates). (**B, C**) Length (**B**) and (**C**) symmetry ratio of aggregates in stretched gel incubated with mitomycin C or aphidicolin for 8 days (n = 214 aggregates). (**D**) Fluorescence images of elongated aggregates cultured for 7 days in the re-embedded or floated gel after stretching. Aggregates were co-stained with anti-Ki67 antibody (green), phalloidin (red), and DAPI (blue). Collagen fibers were labeled with mCherry-CNA35 (magenta). (**E**) Percentage of Ki67 positive cells in the aggregates cultured for 7 days in the re-embedded or floated gel after stretching. (n = 56 aggregates). (**F**) Percentage of Ki67-positive cells in the elongating area of aggregates in the re-embedded or floated gel (n = 47 aggregates). All data are means ± SEM; ns, not significant, **p<0.01, ***p<0.001. Data in (**E, F**) were analyzed by unpaired Student's *t*-test. Data in (**A–C**) were analyzed by one-way ANOVA Tukey's multiple comparisons test.

The online version of this article includes the following source data and figure supplement(s) for figure 5:

**Source data 1.** Original data for quantitative analysis in *Figure 5*.

**Figure supplement 1.** Representative images for *Figure 5A*.

However, proliferation was not increased if gels were allowed to float, rather than being re-embedded, after stretching. Floating the gels reduced the proportion of proliferating cells in the aggregates overall (*Figure 5D and E*) as well as those specifically within the area of elongation (*Figure 5F*), compared with gels that had been re-embedded to preserve collagen polarization. This implied that it was sustained polarization of the gel, rather than simply transfer into the collagen environment, that could stimulate cell proliferation to elongate aggregates.

## Polarized collagen promotes cell proliferation via ERK pathway

Extracellular signal-regulated kinase (ERK1/2) is one of the major signals that stimulates cell proliferation in mammary epithelial cells (*Moreno-Layseca and Streuli, 2014*; *Streuli and Akhtar, 2009*; *Walker and Assoian, 2005*). To evaluate its possible role in aggregate elongation, we expressed the ERK/KTR-mClover biosensor, which translocates out of the nucleus when ERK is activated (*de la Cova et al., 2017*). We confirmed the action of the sensor, which accumulated in nuclei as well as in the cytoplasm when monolayer cultures were treated with the ERK inhibitor FR180204 (50 μM) (*Figure 6—figure supplement 1A*). Using this sensor in 3D cultures, we scored cells that showed nuclear exclusion of the sensor as showing activation of ERK. We found that the proportion of ERK-activated cells was significantly greater in the elongating regions of aggregates that had spontaneously broken symmetry than in the non-elongating areas (*Figure 6A and B*). This was especially apparent in cells at the surface of aggregates that were in contact with the collagen gel (*Figure 6C*). In contrast, although Hippo pathway signaling has been implicated in breaking the symmetry of epithelial organoids (*Serra et al., 2019*), we did not detect any changes in nuclear Yap1 staining that might indicate that this pathway was being activated in our experiments (*Figure 6—figure supplement 1B and C*). This suggested that ERK 1/2 signaling might be a candidate for collagen polarization to stimulate cell proliferation.

This notion was supported by finding that FR180204 (50 μM) reduced cell proliferation in response to stretch polarization (*Figure 6D and E*) and decreased the proportion of aggregates that elongated (*Figure 6F*). As well, the length (*Figure 6G*) and asymmetric morphology (*Figure 6H*) were reduced in those aggregates that did elongate. Thus, ERK1/2 appeared to be critical for collagen polarization to stimulate cell proliferation for aggregate elongation.

## Integrins are necessary for polarized collagen to stimulate elongation

Integrins are the major ECM receptors in the mammary gland and other epithelia, and they control diverse aspects of cellular function, including cell proliferation (*Miranti and Brugge, 2002*; *Wozniak et al., 2003*). We therefore asked if integrin signaling might have stimulated cell proliferation and aggregate elongation in response to collagen polarization (*Figure 7A*).

We focused on the β1-integrins that have been implicated in regulating cell proliferation during mammary gland development (*Li et al., 2005*). This class of integrins contains a number of collagen 1 receptors, including α2β1-integrin (*Heino, 2000*; *Käpylä et al., 2000*), which was expressed in our MCF10A cells (*Figure 7B*). Then we used the inhibitory AIIB2 antibody to block β1-integrin during stretch polarization experiments. mAb AIIB2 (15 μg/ml) was added as aggregates were transplanted into the collagen gel rings and replenished after 7 days. mAb AIIB2 blocked the induction of proliferation in stretch-polarized gels, the proportion of Ki67-positive cells being significantly reduced in AIIB2-treated cultures compared with controls (*Figure 7C and D*). This was accompanied by inhibition of elongation. After 3 days incubation, 49.19% ± 1.61% aggregates started to elongate in controls, whereas less than 20% of aggregates elongated when β1-integrins were inhibited (AIIB2: 17.86% ± 4.78%) (*Figure 7E*). Moreover, even when mAb AIIB2-treated aggregates eventually elongated upon prolonged culture (comparing the proportion of elongated cells at 7 days with that at 3 days), these aggregates remained shorter (*Figure 7F*) and less asymmetric (*Figure 7G*) than in the absence of AIIB2. Finally, we asked if the ERK response required β1-integrins. Indeed, we found that the ability of stretch polarization to stimulate ERK signaling at early (day 2) and later stages (day 7) was reduced when β1-integrins were blocked with mAb AIIB2 (*Figure 7H and I*). This implied that a β1-integrin-ERK pathway was responsible for stimulating epithelial proliferation.

## Discussion

These results lead us to conclude that a polarized collagen 1 matrix can promote epithelial elongation by stimulating locoregional cell proliferation. We infer this because (1) locoregional cell proliferation was a striking feature of MCF10A elongation in our experiments, whose loss was not compensated for by other morphogenetic processes. (2) The pattern of proliferation within aggregates was critically influenced by the polarized organization of the surrounding collagen 1 matrix. Together, we suggest that these reveal an interplay between collagen polarization and cell proliferation that can guide epithelial elongation (*Figure 8*).

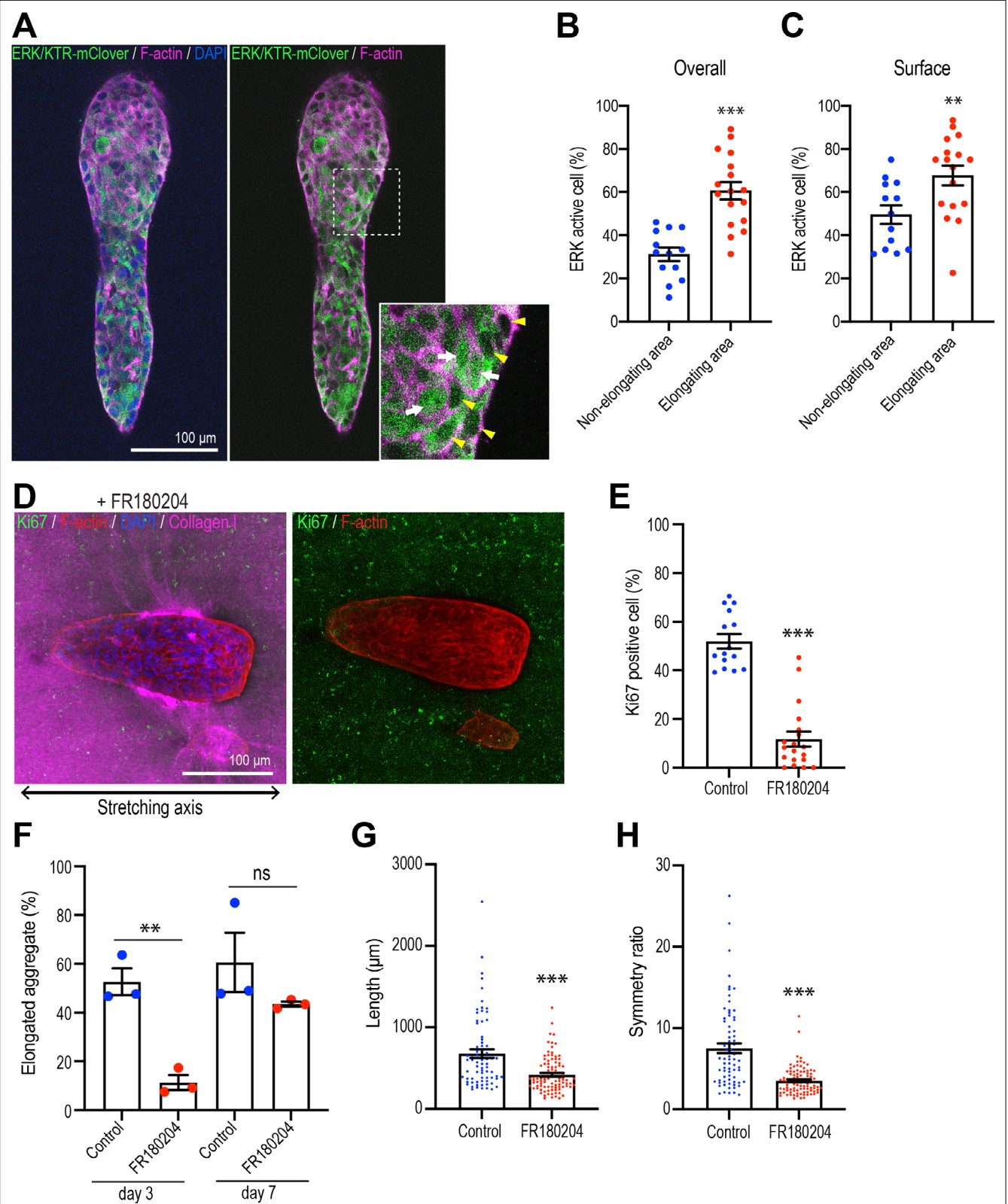

**Figure 6.** Polarized collagen promotes cell proliferation via the ERK pathway. (**A**) Fluorescent images of aggregates expressing ERK/KTR-mClover biosensor cultured for 7 days and stained with phalloidin (magenta) and DAPI (blue). Cells that showed nuclear exclusion of the biosensor (yellow arrowheads) were scored as ERK-activated, while cells that showed both nuclear and cytoplasmic localization of the sensor (white arrows) were scored as inactive. (**B**) Percentage of ERK active cells in elongating areas and non-elongating areas of the aggregates (n = 30 aggregates). (**C**) Percentage

*Figure 6 continued on next page*

*Figure 6 continued*

of ERK active cells at the surface in elongating and non-elongating areas of aggregates (n = 30 aggregates). (**D**) Fluorescent images of aggregates cultured for 7 days treated with FR180207 after gel stretching. Aggregates were co-stained with anti-Ki67 antibody (green), phalloidin (red), and DAPI (blue). Collagen fibers were labeled with mCherry-CNA35 (magenta). (**E**) Percentage of Ki67-positive cells in aggregates incubated with FR180207 for 7 days after stretching (n = 24 aggregates). (**F**) Effect of inhibiting ERK on stretch-induced aggregate elongation. Proportion of elongated aggregates in stretched gel incubated with FR180207 for 3 days and 7 days (N = 3 independent experiments). (**G**) Length and (**H**) symmetry ratio of elongated aggregates incubated with FR180207 for 7 days (n = 166 aggregates). All data are means ± SEM; ns, not significant, \*\*p<0.01, \*\*\*p<0.001. Data in (**B, C, E, H**) were analyzed by unpaired Student's *t*-test.

The online version of this article includes the following source data and figure supplement(s) for figure 6:

**Source data 1.** Original data for quantitative analysis in *Figure 6*.

**Figure supplement 1.** ERK biosensor and YAP1 localization in MCF10A cells.

Cell proliferation was a distinguishing feature of the elongation process in our MCF10A model. Increased proliferation was observed in elongating aggregates and, indeed, began to increase before elongation was detected. Furthermore, proliferation was greater in the regions of the aggregates that underwent elongation compared to those areas that did not. This suggested that a locoregional increase in cell proliferation might be important for the elongation process, a notion that was supported when elongation was inhibited by blocking proliferation. Elongation was reduced when mitomycin C and aphidicolin were added from the beginning of the assays, consistent with earlier evidence that inhibiting proliferation early in the culture process blocked branching morphogenesis in mouse mammary organoids (*Ewald et al., 2008*). We also found that elongation was compromised if proliferation was inhibited after the elongation process had begun, a later dependence that was not seen in organoids (*Huebner et al., 2016*). Differences in cell system may therefore influence the impact of cell proliferation on elongation. Nonetheless, our data strongly suggest that locoregional stimulation of cell proliferation was required for epithelial elongation in our model.

This raised the question of how cell proliferation might be preferentially increased within specific regions of MCF10A aggregates. Several of our observations implicate polarized reorganization of the collagen 1 matrix in this specification process. First, locoregional polarization of the collagen 1 matrix accompanied aggregate elongation when cell aggregates spontaneously broke symmetry. This reorganization was distinguished by increased coherency of the matrix, consistent with an increase in collagen bundling, as well as polarization of the gel so that it became oriented with the axis of the elongating aggregate. Matrix reorganization concentrated where the aggregates were elongating and proliferating, as has been reported earlier (*Brownfield et al., 2013*; *Buchmann et al., 2021*; *Gjorevski et al., 2015*). Second, aggregate elongation was stimulated when the gel was polarized by applying an external stretch. Similarly, collagen 1 orientation has been reported to induce mammary organoid branching (*Brownfield et al., 2013*). But the stimulatory effect on elongation was lost if the stretched gels were allowed to lose their polarization or if aggregates were transplanted into a naïve, unstretched gel. This implied that the cells were responding to the altered organization of the gel, rather than simply exposure to collagen 1. Consistent with what we had observed when aggregates spontaneously broke symmetry, stretch-induced polarization of gels stimulated cell proliferation, and this was necessary for the accelerated elongation process to occur. This appeared to be mediated by a β1-integrin and ERK-dependent pathway.

What property of the polarized collagen 1 was being recognized by the cells to enhance their proliferation? The polarization of collagen networks can have complex effects on cells. Increasing fibril alignment in acellular collagen gels was reported to increase fiber stiffness and decrease pore size (*Riching et al., 2014*; *Taufalele et al., 2019*), even in the absence of externally applied forces. Moreover, in single-cell cultures aligned collagen fibers were associated with larger focal adhesions than non-aligned fibrils (*Doyle and Yamada, 2016*). One possibility, then, is that integrin signaling was responding to the increased stiffness of the polarized collagen gels. This is consistent with the well-characterized role for integrins to sense changes in matrix stiffness (*Giannone and Sheetz, 2006*). As well, collagen 1 can bind to, and sequester, a variety of growth factors. Whether this reservoir can be released when cells apply tension to collagen is an interesting question for further consideration (*Wipff and Hinz, 2008*).

Cell migration is a key driver of branching morphogenesis (*Gjorevski et al., 2015*; *Huebner et al., 2016*), and earlier experiments reported that the application of external stretch to collagen-embedded

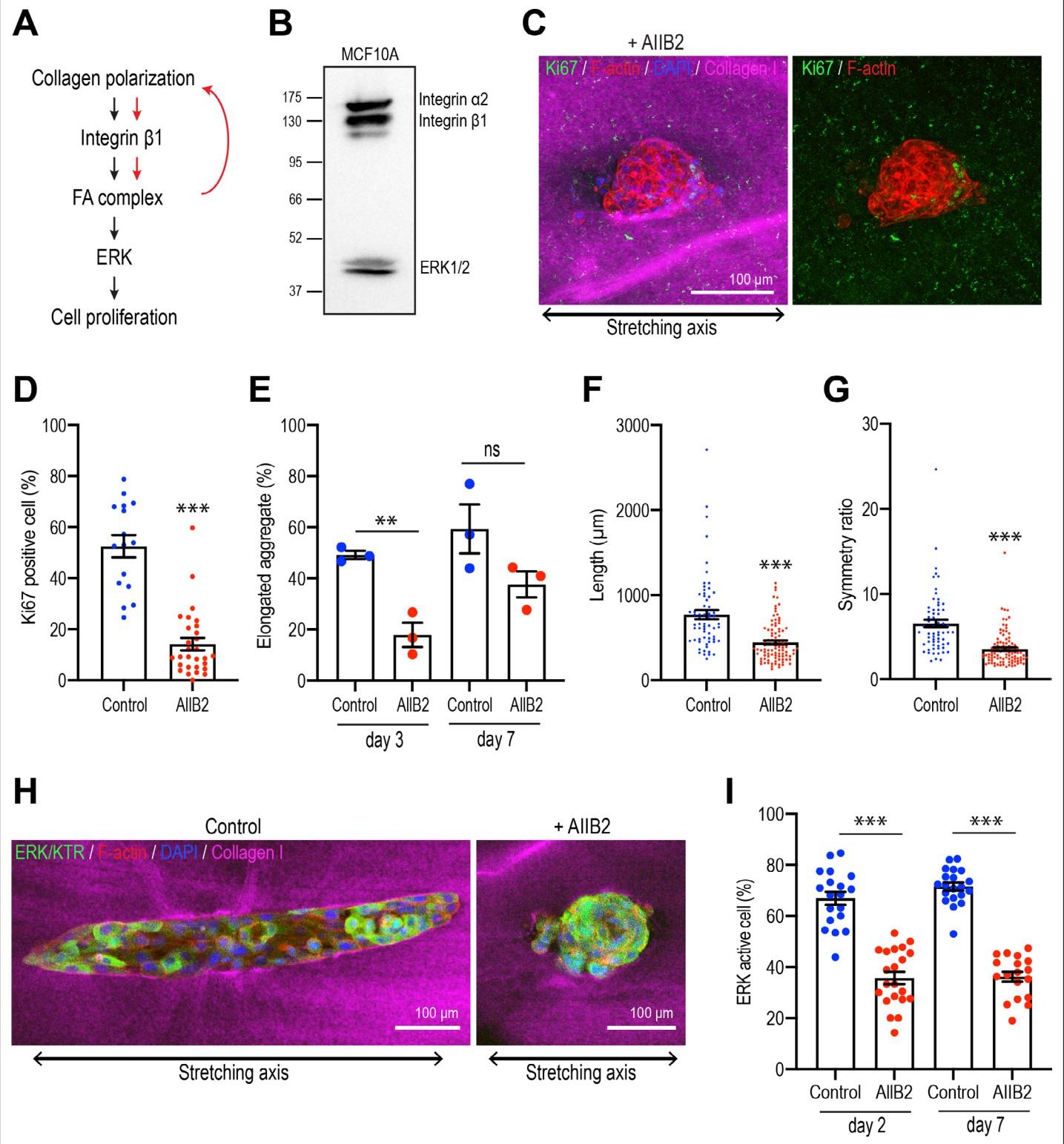

**Figure 7.** Integrins are necessary for polarized collagen to stimulate elongation. (**A**) Schematic of potential integrin-ERK pathway that mediates the effect of collagen polarization on cell proliferation. (**B**) Immunoblot of integrin α2, β1, and ERK1/2 protein levels in MCF10A cell lysate. (**C**) Fluorescent images of aggregates cultured for 7 days treated with AIIB2 antibody after gel stretching. Aggregates were co-stained with anti-Ki67 antibody (green), phalloidin (red), and DAPI (blue). Collagen fibers were labeled with mCherry-CNA35 (magenta). (**D**) Percentage of Ki67-positive cells in the aggregates incubated with AIIB2 antibody for 7 days after stretching the gels (n = 35 aggregates). (**E**) Proportion of elongated aggregates in stretched gel incubated with AIIB2 antibody for 3 days and 7 days (N = 3 independent experiments). (**F**) Length and (**G**) symmetry ratio of elongated aggregates incubated

*Figure 7 continued on next page*

*Figure 7 continued*

with AIIB2 antibody for 7 days (n = 134 aggregates). (**H**) Fluorescent images of aggregates expressed with ERK/KTR-mClover and incubated with IgG (control) or AIIB2 antibody for 2 days after gel stretching. Aggregates were co-stained with phalloidin (red) and DAPI (blue). Collagen fibers were labeled with mCherry-CNA35 (magenta). (**I**) Percentage of ERK active cells in aggregates incubated with AIIB2 antibody for 2 days or 7 days in stretched gel (n = 78 aggregates). All data are means ± SEM; ns, not significant, **p<0.01, ***p<0.001. Data in (**D–G, I**) were analyzed by unpaired Student's *t*-test.

The online version of this article includes the following source data for figure 7:

**Source data 1.** Original data for quantitative analysis in *Figure 7*.

**Source data 2.** Original image of western blot in *Figure 7B*.

aggregates could promote elongation through cell migration (*Brownfield et al., 2013*). We, too, observed cell migration in our aggregates and, although their speeds were not different, migrating cells within elongating parts of the aggregates tended to align with the axis of elongation and move slightly more persistently than cells moving within the non-elongating regions of the aggregates. These features of collective migration might be expected to contribute to aggregate elongation.

Interestingly, blocking cell proliferation decreased the apparent persistence of the migrating cells and their alignment with the principal axes of the aggregates. Therefore, rather than collective migration being enhanced to compensate for the decrease in proliferation, cell proliferation appeared to support these elongation-facilitating aspects of cell migration. One possible explanation is that persistence and directional alignment were being guided by the reorganized collagen that, in effect, served to create a 3D micropattern around the aggregates. Studies in 2D systems have shown that cells can orient their patterns of migration when collagen networks condense and form bundles (*Mohammed et al., 2020*; *Wang et al., 2018*). As collagen reorganization was reduced by inhibiting proliferation, mitomycin C and aphidicolin might have affected cell migration indirectly through matrix organization. Such micropatterning might also influence other aspects of cell behavior during elongation. Micropatterning experiments using 2D substrata have shown that anisotropic confinement can orient patterns of cell division (*Théry et al., 2005*), potentially by orienting tensile stresses (*Campinho et al., 2013*; *Legoff et al., 2013*), as would also be predicted to accompany stiffer, condensed collagen bundles. Consistent with this, we observed that cells within the elongating areas appeared to orient their divisions with the axis of elongation, an effect that would also be predicted to enhance elongation. These observations reinforce the notion that interplay between cell proliferation and collagen (re)organization is a key contributor to epithelial elongation.

The reductionist model that we used in these experiments allowed us to focus on testing how collagen organization could promote epithelial elongation. How might the results from this system operate in the more complex environment of the tissue? One important factor to consider is the basement membrane, which can separate the epithelial cell compartment from collagen 1 in the stroma. However, earlier studies reported that the basement membrane thins substantially and is remodeled in regions of epithelial elongation, such as the tips of mammary or salivary gland buds (*Harunaga et al., 2014*; *Silberstein and Daniel, 1982*; *Williams and Daniel, 1983*), potentially allowing elongating regions of epithelia to engage with collagen 1. Evaluating how cell proliferation and stromal collagen organization are coordinated with other components of the ECM in physiological models of developing glands will be an important question for future research.

In conclusion, we propose the following working model (*Figure 8*): regional polarization of the collagen 1 matrix begins in response to anisotropies in force that aggregates exert upon their ECM via their integrin adhesions (*Figure 7A*). Cell-based forces exerted through integrins can apply strain on collagen fibrils (*Brownfield et al., 2013*; *Buchmann et al., 2021*; *Gjorevski et al., 2015*; *Hall et al., 2016*) and, consistent with this, collagen reorganization in our experiments was compromised by inhibiting cellular contractility. However, compression of collagen by increasing cell numbers or cell movements could also have contributed (*Buchmann et al., 2021*). By implication, collagen polarization will be greater around the parts of the aggregate that are generating more force. The consequent locoregional collagen polarization then stimulates further cell proliferation in the adjacent parts of the aggregate to sustain elongation. Biochemical contact with collagen may provide a basal stimulus for cell proliferation, which is further enhanced where collagen becomes polarized, to provide a boundary condition that promotes elongation. It was interesting to note that collagen reorganization was compromised when cell proliferation was blocked from the outset of the assays. Possibly, subtle differences in cell proliferation may have contributed to generating initial anisotropies in force to

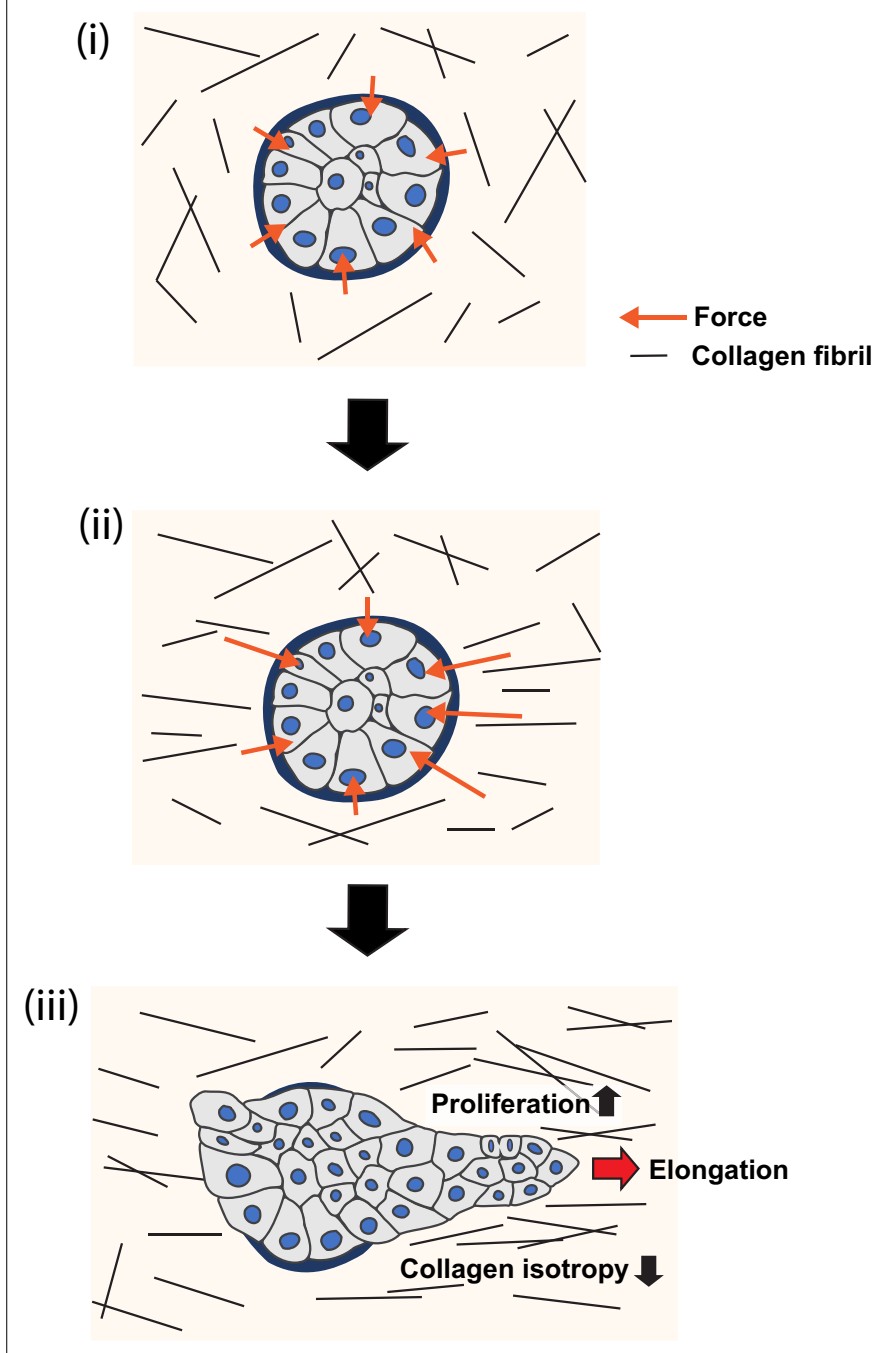

**Figure 8.** Model of collagen polarization as a structural memory for epithelial anlage elongation. (**i**) Initially isotropic epithelia anlage exert isotropic patterns of force on a non-polarized collagen 1 gel. (**ii**) Initial anisotropies in force associated with symmetry breaking of the aggregate exert strain on collagen fibrils leading to bundling and polarization. (**iii**) The polarized collagen matrix provides a structural memory that promotes regional cell proliferation to direct further elongation of the anlage.

reorganize the gels. If so, we speculate that the increase in cell proliferation may have contributed to further collagen reorganization, with the capacity to develop a feedback system that might help sustain aggregate elongation. This would allow polarized collagen networks to effectively provide a structural memory of the initial axis of symmetry breaking, that is, a relatively long-lived spatial cue that directs further elongation of the epithelial anlagen.

# Materials and methods

## Cell culture and lentivirus infection

MCF10A human mammary epithelial cells (CRL-10317) and HEK293T human embryonic kidney 293 cells (CRL-3216) were obtained from ATCC. Both cells were confirmed mycoplasma negative by PCR. MCF10A cells were cultured in DMEM/F12 medium supplemented with 5% horse serum, 10 μg/ml insulin, 0.5 μg/ml hydrocortisone, 100 ng/ml cholera toxin, 20 ng/ml EGF, 100 units/ml Penicillin and 100 units/ml Streptomycin as previously described (*Debnath et al., 2002*).

HEK-293T cells were transfected using Lipofectamine 2000 (Invitrogen) for lentiviral expression vector pLL5.0 and third-generation packaging constructs pMDLg/pRRE, RSV-Rev, and pMD.G. Third-generation packaging constructs were kindly provided by Prof. James Bear (UNC Chapel Hill, NC). After transducing MCF10A cells with either pLL5.0-EGFP-HRasC20, pLL5.0-NLS-mCherry, or pLenti-ERK/KTR-mClover lentivirus construct, we isolated highly-expressing cells using fluorescence-activated cell sorting (Influx Cell sorter; Cytopeia).

## 3D Matrigel and collagen culture

MCF10A aggregates and acini were cultured on 100% growth factor-reduced Matrigel (FAL354230; Corning) and overlayed with Matrigel including medium in eight-well chamber coverglass (0030742036; Eppendorf) as previously described (*Debnath et al., 2002*; *Debnath et al., 2003*). In brief, single isolated MCF10A cells were mixed with an assay medium containing 2% Matrigel, 5 ng/ml EGF, and seeded on a solidified layer of 100% Matrigel at 10,000 cells/well. After seeding cells on Matrigel, medium was changed every 4 days.

In collagen culture, single isolated MCF10A cells were mixed with collagen gel at 20,000 cells/well (FAL354236; Corning). This mixture was neutralized with NaOH and HEPES buffer and collagen concentration was adjusted to 1.0 mg/ml with culture medium on ice. The pH of collagen solution was checked by litmus paper. Collagen solution with cells was seeded in eight-well chamber coverglass and solidified at 37°C for 30 min.

## Transplantation of MCF10A aggregates

MCF10A aggregates cultured on Matrigel for 10 days were washed with PBS and incubated with cold cell recovery solution (FAL354253; Corning) for 30 min on ice. Aggregates were collected into a tube containing the cell recovery solution, spun down at 1200 rpm, and washed with cold PBS. PBS-washed aggregates were resuspended in MCF10A culture medium and mixed with collagen solution neutralized with NaOH and HEPES buffer. Collagen gel solution with aggregates was solidified at 37°C for 30 min.

Aggregates embedded in collagen gel were isolated from gel by dissolving the gel with collagenase (C2799; Sigma). Gels were washed with PBS and incubated in 20 μg/ml collagenase in Hanks' Balanced Salt Solution (H8264; Sigma) at 37°C for 30 min to dissolve the collagen gel. Aggregates were collected with incubated solution into the tube and mixed with DMEM/F12 medium, which contains 20% horse serum. Solution was spun down at 1200 rpm and pellet was washed with PBS. PBS-washed aggregates were resuspended in culture medium and re-embedded in collagen gel.

Plasmids pLL5.0-NLS-mCherry was constructed in a previous study (*Leerberg et al., 2014*). pLent-ERK/KTR-mClover was obtained from Addgene (#59150). pLL5.0-EGFP-HRasC20 was generated by insertion of EGFP-HRasC20 fragment that was amplified from pEGFP-F (#6074-1; Clontech) by PCR into pLL5.0 vector. pLL5.0-EGFP (gift from Prof. James Bear; UNC Chapel Hill) was digested with EcoRI and SbfI to remove EGFP, and then EGFP-C20 fragment was inserted and ligated by In-Fusion cloning kit (638910; Clontech).

## Antibodies and inhibitors

Primary antibodies used for immunocytochemistry in this study were rabbit anti-Ki67 (ab15580; Abcam), mouse anti-laminin V (MAB19562; Chemicon), rat anti-E-cadherin (131900; Invitrogen), mouse anti-GM130 (610822; BD), rabbit anti-fibronectin (F3648; Sigma), and rabbit anti-YAP1 (4912; Cell Signaling Technology). F-actin was stained with AlexaFluor 488-, 594-, 647-phalloidin (Invitrogen). Primary antibodies used for immunoblot were rabbit anti-integrin α2 (ab181548; Abcam), mouse anti-integrin β1 (610467; BD Transduction Laboratories), and rabbit anti-p44/42 MAPK (Erk1/2) (9102; Cell

Signaling Technology). AIIB2 antibody for blocking integrin β1 was purchased from Developmental Studies Hybridoma Bank and treated at 15 µg/ml in this study.

Proliferation inhibitors mitomycin C (M7949; Sigma) and aphidicolin (178273; Merck) were used at 10 µM and 2 µM, respectively. Myosin II inhibitor blebbistatin (203390; Sigma) and ROCK inhibitor Y-27632 (688000; Sigma) were used at 25 µM and 30 µM, respectively. ERK1/2 inhibitor FR180204 (SC-203945; Santa Cruz) was used at 50 µM. Except for mitomycin C, cell aggregates were treated with the inhibitors after transplantation into collagen gels. Inhibitors were then replenished when medium was changed and left in the collagen culture until cell aggregates were ready to be fixed. For mitomycin C washout experiments, cell aggregates were treated with mitomycin C for 1 hr, then washed three times with PBS, and left to recover in MCF10A culture medium.

## Immunocytochemistry and microscopy

3D cultured cells in gel were fixed with 4% paraformaldehyde in cytoskeleton stabilization buffer (10 mM PIPES at pH 6.8, 100 mM KCl, 300 mM sucrose, 2 mM EGTA, and 2 mM $MgCl_2$) at room temperature for 30 min, followed by a treatment for 30 min with 0.5% TritonX-100 and 10% goat serum in PBS for 1 hr at room temperature. Then they were stained with primary antibodies for overnight at 4°C. Subsequently, cells were washed with PBS and incubated with secondary antibodies with phalloidin and DAPI for 1 hr at room temperature.

Confocal images were acquired with an upright Meta laser scanning confocal microscope (LSM710; Zeiss) equipped with plan-Apochromat 20× 0.8 NA or 40× 1.3 NA objectives (Zeiss) and zen2012 software (Zeiss). The fluorescent images of collagen fibrils probed with mCherry-CNA35 were acquired by inverted microscope (Ti2; Nikon) equipped with Dragonfly spinning disc (Andor) by using plan-Apo 10× 0.45 NA, 20× 0.75 NA, or 40× 0.95 NA dry objectives (Nikon) and Fusion software (Andor).

Fluorescent images of the SHG signal from collagen 1 were collected with an inverted confocal microscope (LSM710; Zeiss) equipped with multiphoton laser by using 860 nm excitation with SHG signal obtained with 690 nm bandpass filter. A 40× 1.3 NA or 63× 1.40 NA plan-apochromat oil objectives (Zeiss) were used to obtain SHG signals.

Fluorescence and phase-contrast live images of elongating aggregates were acquired with an inverted fluorescence microscope (IX81; Olympus) equipped with CCD-camera (Hamamatsu) and an incubation box (Clear State Solutions) maintained at 37°C and 5% $CO_2$ with gas controller (OkoLab), using plan-Apo 10× 0.4 NA objective (Olympus) and CellSens software (Olympus).

## Collagen gel labeling and stretching

Collagen fibrils were labeled with mCherry-CNA35 by mixing with purified mCherry-CNA35 protein at 2 µM before gelling. Gels were solidified at 37°C for 30 min.

PDMS gel frame and stretchers for gel stretching were kindly gifted from Dr. James Hudson (QIMR Berghofer, Queensland, Australia). 1.5 mg/ml collagen gels were solidified as a ring-shape in PDMS frame. PDMS stretchers were inserted into the hole of the gel and expanded, and incubated for 4 hr in culture medium. Stretched gels were released and then floated in medium or re-embedded in collagen gel. For re-embedding, stretched gels were put in 1.5 mg/ml collagen gel and solidified at 37°C for 30 min.

## CNA35-mCherry protein purification

Protein expression vector pET28a-mCherry-CNA35 was obtained from Addgene (#61607). Transformed *Escherichia coli* (BL21) was cultured in 400 ml LB, and induced protein expression with isopropyl β-D-1-tthiogalactopyranoside for 20 hr at 25°C. Cultured bacteria were spun down, and collected bacteria pellet was sonicated. The lysate was centrifuged, and the resulting supernatant was run through a column filled with N-NTA His-band resin (Millipore). Bound protein was eluted and then dialyzed for overnight in PBS at 4°C. Endotoxin was removed by endotoxin removal columns (88274; Thermo) following with manufacturer's protocol.

## Quantitative analysis of collagen fibril alignment and aggregates elongation

Fluorescent confocal images of collagen fibrils acquired SHG microcopy or CNA35 probes were used for collagen fibril analysis. We analyzed collagen organization in these images using Orientation J, which calculates the local orientation and isotropy for each pixel in an image based on the structure tensor for that pixel (*Rezakhaniha et al., 2012*). The structure tensor is evaluated for each pixel of the given image by calculating the spatial partial derivatives by using (a cubic B-spline) interpolation. The local orientation and isotropy for each pixel are computed based on the eigenvalues and eigenvectors of the structure tensor. We characterized three features in the organization of collagen fibrils orientations (*Clemons et al., 2018*): (i) the coherency of fibrils, defined as co-orientation in the same direction, as a measure of bundling; (ii) isotropy, the distribution of fibril orientation in the field of analysis; and (iii) polarization, defined as the principal axis of fibril orientation in anisotropic gels, relative to a reference axis. 50 × 50 µm or 100 × 100 µm size of ROI were selected from SHG images or CNA35 probed images, respectively, and used for analysis. Elongation axis of aggregates was measured by the angle tool in ImageJ, and then we calculated the angle difference between principal axis of collagen fibrils or stretching axis. The fold difference of gel coherency was measured by dividing the coherency of fibrils in the elongating area by the non-elongating area.

The symmetry ratio of aggregates was measured by dividing the longest length by widest width of aggregates.

## Quantitative analysis of proliferating cells and nuclear division angle

The number of cells co-stained with DAPI and anti-Ki67 antibody was counted by Imaris software (Bitplane). The number of Ki67-positive cells was divided by the total number of nucleus stained with DAPI to calculate percentage of Ki67-positive cells. For live imaging, cells were expressed with NLS-mCherry to count their number. The frequency and orientation of cell division were analyzed from the time-lapse images by ImageJ. We set the elongation axis of aggregate as a reference and measured the angle difference between elongation axis and dividing axis of nucleus.

## Analysis of nuclear tracking

To obtain the migration speed of cells, we tracked individual cell nuclei using the Spot function in the Imaris software (*Huebner et al., 2016*). The fluorescent images of cells expressing NLS-mCherry were acquired every 10 min for at least 50 hr. To obtain the track displacement angle, we first calculated the displacement angle of nuclei from the displacement of X and Y axis between first and last position, and then measured the angle difference from elongating angle of aggregates. Track straightness was calculated from track displacement by track length. Inhibitors were treated 1 hr before the imaging.

## Quantitative analysis of ERK/KTR biosensor

ERK activity was judged by the location of mClover fluorescent signal in individual cells. Briefly, fluorescent-tagged KTRs translocate between nuclei and cytoplasm depends on kinase activity. When ERK activity is high, KTR-mClover should localize in cytoplasm (*de la Cova et al., 2017*). Aggregates expressing ERK/KTR-mClover were transplanted into collagen gel and were co-stained with phalloidin and DAPI after fixation. To judge the delocalization of KTR-mClover, we used line intensity scan in single-plane images and then manually scored ERK active cells as those where the fluorescent signal was excluded from the nucleus (compared with cells that showed both nuclear and cytoplasmic localization, which were scored as inactive).

## Statistical analysis

Significance was determined by unpaired Student's *t*-test and one-way ANOVA by using GraphPad Prism 8 (GraphPad software).

## Acknowledgements

We thank Selwin Wu for advice and all our lab colleagues for their feedback and support. The authors were supported by the National Health and Medical Research Council of Australia (Fellowship 1136592 and GNT1123816, 1140090 to ASY), Australian Research Council (DP19010287, 190102230 to AY and FT190100516 to SS), a Snow Medical Fellowship to JH, and a postdoctoral fellowship from The Uehara Memorial Foundation to HK.

## Additional information

### Funding

| Funder | Grant reference number | Author |
| --- | --- | --- |
| National Health and Medical Research Council | Fellowship 1136592 | Alpha S Yap |
| National Health and Medical Research Council | GNT1123816 | Alpha S Yap |
| National Health and Medical Research Council | 1140090 | Alpha S Yap |
| Australian Research Council | DP19010287 | Alpha S Yap |
| Australian Research Council | 190102230 | Alpha S Yap |
| Australian Research Council | FT190100516 | Samantha J Stehbens |
| Snow Medical Fellowship | | James Hudson |
| Uehara Memorial Foundation | Postdoctoral fellowship | Hiroko Katsuno-Kambe |

The funders had no role in study design, data collection and interpretation, or the decision to submit the work for publication.

### Author contributions

Hiroko Katsuno-Kambe, Conceptualization, Formal analysis, Investigation, Methodology, Project administration, Validation, Writing – original draft; Jessica L Teo, Conceptualization, Writing – review and editing; Robert J Ju, Methodology; James Hudson, Methodology, Writing – review and editing; Samantha J Stehbens, Writing – review and editing; Alpha S Yap, Project administration, Supervision, Writing – original draft, Writing – review and editing

### Author ORCIDs

Hiroko Katsuno-Kambe  http://orcid.org/0000-0003-0292-3506
Robert J Ju  http://orcid.org/0000-0002-9850-9803
Samantha J Stehbens  http://orcid.org/0000-0002-8145-2708
Alpha S Yap  http://orcid.org/0000-0002-1038-8956

### Decision letter and Author response

Decision letter https://doi.org/10.7554/eLife.67915.sa1
Author response https://doi.org/10.7554/eLife.67915.sa2

## Additional files

### Supplementary files

• Transparent reporting form

### Data availability

All data generated and analysed in this study are included in the manuscript and supporting files. Source data files have been provided for Figure 1-7.

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
