## [Decision Letter]

**Acceptance summary:**

This paper makes use of elegant experiments to explore the role of collagen matrix polarization during symmetry break and elongation of three-dimensional epithelial anlagen. Polarized collagen stimulates regional cell proliferation through β1 integrin and ERK signaling which is necessary to drive symmetry break and elongation. These findings will be of interest to a broad audience of developmental and cell biologists.

**Decision letter after peer review:**

Thank you for submitting your article "Collagen polarization provides a structural memory for the elongation of epithelial anlage" for consideration by *eLife*. Your article has been reviewed by 3 peer reviewers, including Matthew Kutys as the Reviewing Editor and Reviewer #1, and the evaluation has been overseen by Jonathan Cooper as the Senior Editor.

Essential revisions:

In this paper, the authors used reductionist models to investigate the interplay between collagen extracellular matrix and the polarized elongation of mammary epithelial anlage and provide new mechanistic insights into this process, which are potentially of broad interest. However, all three reviewers felt that there are several issues that need to be addressed before this paper can be published, most significantly those summarized below:

1) All reviewers felt the data did not fully support the conclusions in several areas. Most notably, there were issues with the suggested causality of proliferation in driving analage elongation and the instructive role of polarized collagen in directing outgrowth that must be addressed.

2) The authors distinguish their work by focusing on elongation rather than branching events, however the approaches and observations are similar to previous studies of branching 3D mammary explant organoids in aligned collagen matrix. How the conclusions of this work, including the observed morphogenic behavior and underlying control mechanisms, advance our conceptual understanding must be made clear.

3) Demonstrating the extent to which the findings from this work apply to mammary tubulogenesis in vivo is required.

*Reviewer #1:*

Katsuno-Kambe and colleagues investigate the mechanism by which non-polarized MCF10A spheroids can be induced to extend multicellular aggregates when cultured within 3D type I collagen extracellular matrix hydrogels. Transplanting preformed aggregates into collagen is sufficient to induce asymmetric elongation of the spheroid. Branching and elongation depends on region-specific cell proliferation and multicellular outgrowths are characterized by the local accumulation of collagen matrix after the initial symmetry break. 3D hydrogel stretching and preformed aligned collagen extracellular matrix is sufficient to direct elongated outgrowth, which is dependent on proliferation, β1 integrin and ERK signaling. While the work is quantitative and the imaging is of high quality, chemical and physical factors that affect the long axis bias of the developing mammary epithelium have been intensively studied in vivo and in vitro, and it is not clear that this work offers a significant conceptual advance. The study also would need validation of observations from their engineered in vitro systems with relevant aspects of in vivo conditions.

Methods and conclusions from this work are similar to those previously reported by the Bissell group (Brownfield et al. Current Biology, 2013). In this study, unpolarized primary murine mammary explants were embedded within 3D collagen hydrogels and axially stretched using a deformable PDMS-based device. Collagen stimulated explant branching and outgrowth, which required collagen fiber polarization. The authors do reference this study, but argue its focus is on branching rather than elongation. However, given the nearly identical methods, analyses, and morphogenic conclusions, it is unclear how this present work is distinct. Similarly, it is unclear if the mechanisms derived from the reductionist system here are relevant to mammary development/tubulogenesis in vivo.

1) Recent and previous work from the Bissell and Nelson groups have explored the role of collagen fiber polarization and/or axial alignment during patterned elongation of the developing mammary gland. While the role collagen still remains poorly understood, it will be important to reconcile these studies with the work presented here, namely by demonstrating the relevance of the findings from this work to mammary tubulogenesis in vivo.

2) Methods and conclusions from this work are similar to those reported by the Bissell group (Brownfield et al. Current Biology, 2013). In this study, unpolarized primary murine mammary explants were embedded within 3D collagen hydrogels and axially stretched using a deformable PDMS-based device. Collagen stimulated explant branching and outgrowth, which required collagen fiber polarization. The authors do reference this study but argue its focus is on branching rather than elongation. However, given the nearly identical methods, analyses, and morphogenic conclusions, it is unclear how this present work is distinct.

3) "In contrast, Rac1 inhibitor treatment did not affect aggregate elongation and collagen fibril rearrangement (data not shown)". These data should be shown, as it conflicts with the conclusions of the Brownfield study. It should be noted that the Rac1 inhibitor NSC23766 only blocks a subset of Rac1 interactions with Rac1GEFs.

4) From Figure 2A, is the gel coherency metric capturing fibril alignment, or the localized accumulation of collagen fluorescent intensity? Is collagen accumulation a function of cells pushing/expanding out into the matrix, or from active fiber recruitment to the outgrowth surface?

What determines whether localized collagen intensity leads to branching and outgrowth? In Figure 2A, there are two distinct regions of polarized collagen, but only one contains outgrowth. There is an overall variability in the brightness, contrast, and resolution of the representative collagen images, making it difficult to interpret across experiments. For instance, polarized collagen appears to be present in the blebbistatin treated outgrowths, however it is difficult to discern due to the different magnification and non-uniform hydrogel intensity.

5) Is proliferation truly driving this behavior? In the representative images (Figure 1C, 1H) it appears that all peripheral MCF10A making basal matrix contact are Ki67 positive. It is a bit difficult to tell when proliferation arresting inhibitors were added in the evolution of these processes. Does acute addition of mitomycin halt, or regress, established outgrowths?

6) Similarly, do stretch-induced elongations regress after removal from stretch and transfer to floating culture?

7) AIIB2 implicates β1 integrin, but not α2β1. The authors should check to see if fibronectin deposition is stimulated by stretch, and is fibronectin present at stochastic outgrowths in non-patterned collagen?

8) "This regional patterning would be necessary if proliferation drives elongation, as a global increase in cell number would cause aggregates to expand isotropically." It is important to acknowledge that a critical regulatory feature missing from this model is a developed basement membrane. in vivo, the local degradation and remodeling of basement membrane mechanically permit region specific expansion of developing branched tissues, such as in the embryonic salivary gland (Harunaga et al. Dev. Dyn., 2014).

9) "The decision to branch is recognized to be a critical checkpoint that is controlled by developmental signaling." It is unclear what distinction the authors are trying to make. There is robust literature suggesting cell nonautonomous mechanisms regulating branching events, notably localized extracellular matrix deposition and heterotypic cell-cell interactions.

10) The aspect ratios of the images in Figure 5D appear to be off.

*Reviewer #2:*

The formation of mammary gland tubules begins as non-polarized cellular aggregates, called anlage, which elongate as solid multicellular cords that eventually form lumens. Katsuno-Kambe et al. investigate the role of the extracellular matrix during the process of elongation, making use of 3D culture models of mammary epithelial MCF10A cells. These cells have previously been shown (e.g Krause et al., Tissue Eng Part C Methods 2008, and others) to form multicellular aggregates that polarize and eventually form lumens when cultured in Matrigel, and when embedded in Collagen I (a major component of the stromal environment) form elongated, solid cords of non-polarized cells. Therefore, to capture the process by which aggregates break symmetry and become elongated, the authors first seeded MCF10A cells in Matrigel until proliferation arrested (resembling anlage), after which aggregates where isolated and embedded into collagen I gels. The authors show increased proliferation of cells in elongating areas of aggregates, and that these elongated aggregates are not formed when proliferation is inhibited. The authors further show that this elongation is accompanied by collagen reorganization into polarized bundles around elongating areas, and by artificially polarizing collagen by application of external stretch show collagen polarization can increase proliferation and guide the direction of elongation. Finally, the authors demonstrate aggregate elongation requires beta1-integrin adhesion and downstream ERK activity.

The strength of this manuscript is the use of a reductionist approach to investigate the role of collagen alignment in the elongation of epithelial anlage, by transplanting MCF10A cell clusters from matrigel into Collagen I gels, and artificially manipulating collagen polarization by stretching these gels. In addition, the exploration of the underlying mechanism of elongation, requiring proliferation and integrin-ERK signaling, is of potential great interest to understanding the role of cues derived from the extracellular matrix in epithelial morphogenesis.

The main weakness of this study is that the data still insufficiently supports several conclusions, in particular on proliferation being a key driver of anlage elongation, and on the reciprocal interaction between proliferation and collagen polarization during this process. The induction of proliferation is only analyzed 8 days after cell clusters have been transferred to Collagen gels, and thus after symmetry break and elongation have already occured. Analyses of proliferation and collagen polarization during the different stages of symmetry break and elongation, and detailed analyses of the distribution of proliferative cells during these stages, are lacking. This is in particular relevant as also in non-elongated areas, cells that are in contact with Collagen appear to be proliferative. Alternative driver mechanisms, for instance the induction of collective migration, are not fully tested and it remains unclear whether proliferation is merely required for, or truly a driver event of aggregate elongation.

The authors investigate a role for integrin signaling and downstream ERK in the induction of proliferation and aggregate elongation in response to collagen polarization. While an integrin β 1 inhibitory antibody and ERK inhibitor indeed prevents elongation, key unanswered questions are whether this pathway is indeed activated by polarized collagen during elongation (i.e. in selective regions of the cell clusters) and whether it is required for the proliferation response itself.

Despite the elegance of the reductionist approach in MCF10A cells in vitro, it remains unclear to what extent these findings generally apply to mammary tubulogenesis under physiological conditions in vivo, which remains largely undiscussed in the manuscript.

Comments for the authors:

1. The authors conclude that cell proliferation is a key driver of epithelial anlage elongation. However, the induction of proliferation is only analyzed 8 days after transfer of cell clusters to Collagen I, after symmetry break and elongation have taken place. And here, analysis of proliferation is limited to cells in the elongated areas versus the remaining cell cluster. It is essential to include analyses of proliferation during the different stages of elongation (i.e. before and upon initial symmetry break and at different time points of elongation), as well as detailed analyses of the distribution of proliferative cells during these stages. This is in particular relevant as from the images (e.g. Figure 1C) it is apparent that also in the non-elongated areas the cells on the outside of the cell cluster (thus in contact with Collagen I) are proliferating (e.g. Figure 1C), yet do not form elongated strands (which also does not support the authors conclusions that "proliferation is selectively increased in regions of aggregates that were elongating"). Finally, analysis of the induction of proliferation should also include a control in which cell clusters are transferred back to matrigel.

2. Alternative driver mechanisms, in particular cell migration, are limited to analysis of migration speed that is similar in different regions of the aggregate. This does not exclude that induction of directional, collective migration could be the main driving event underlying elongation, which remains unexplored. As such, it remains undetermined whether proliferation is merely required for, or truly a driver event of aggregate elongation.

3. The authors mention in the discussion that "the initial step of symmetry-breaking was associated with reorganization of collagen fibrils along the aggregate (…) which became increasingly bundled and aligned as elongation proceeded". However, data showing these changes in collagen during the different steps are not presented in the manuscript. This is in particular important as the authors conclude that collagen polarization induces proliferation, and therefore the authors should include a more detailed analyses of both processes (collagen reorganization and proliferation) over time after cell clusters are transferred to Collagen gels, and how they are associated with each other in individual cell clusters.

In addition, can the authors explain the induction of proliferation in cells in contact with collagen in the non-elongating areas (e.g. Figure 1C, see comment 1), as this suggest contact with collagen is sufficient to increase proliferation and collagen polarization is not required?

4. In line with the above comment, the conclusion that collagen polarization promotes proliferation is largely based on the experiments in which collagen gels were stretched. However, the effect on proliferation is again only assessed following 8 days of culture in these gels, and thus after aggregate elongation. To support their conclusion, proliferation should be analyzed at early time points following stretching. This would also address whether the earlier symmetry break observed after stretching is a consequence of increased/faster induction of proliferation, or involves the ability of already proliferating cells to drive epithelial elongation (which will affect the interpretation of these data).

5. The authors demonstrate that inhibition of proliferation prevents aggregate elongation. It is difficult to interpret from these experiments whether proliferation could truly be a key driver of elongation, as aggregates appear to remain much smaller following mitomycin C and Aphidicolin treatment (Figure 1H). It should be shown whether in the absence of the inhibitors, symmetry break and elongation is induced at this aggregate size (and thus whether the observed effect represents a direct or indirect of the inhibitors on elongation). In addition, proliferation inhibitors should also be given after the symmetry break has taken place, to test whether this prevents protrusions that are already present from elongating further.

6. The authors investigate a role for integrin signaling and downstream ERK in the induction of proliferation and aggregate elongation in response to collagen polarization. While indeed both integrin beta1 and ERK activity are necessary for aggregate elongation, whether this pathway is directly involved in aggregate elongation remains unclear, and analysis of cell proliferation is lacking. The authors should test with available antibodies whether this pathway is specifically induced by polarized collagen (by analyzing different regions of the aggregate during symmetry break and elongation). In addition, effect of inhibitors on the level and distribution of Ki67+ cells should be shown.

7. To control that effects observed in stretched collagen gels are truly mediated by the induced collagen polarization, the authors allowed gels to float in the medium to revert gel isotropy to that of unstretched gels. While collagen polarization is indeed reverted, cells cultured in floating gels will not able be able to generate stresses themselves (e.g. Halliday et al., 1995). From these experiments it can therefore not be concluded whether it is truly the collagen polarization or stresses generated by the cells that are needed for aggregate elongation.

8. It remains unclear to what extent the findings from this manuscript apply to mammary tubulogenesis in vivo, which remains undiscussed in the manuscript. While some experiments may be difficult to verify in vivo (e.g. to monitor collagen alignment), it would be relevant to at least to know whether in vivo mammary epithelial cells indeed come in contact with collagen I during anlage elongation, and preferably also show whether this is accompanied by a specific proliferation response.

*Reviewer #3:*

Understanding tissue morphogenesis requires an understanding of underlying molecular and mechanical mechanisms. Many tissues form a network of branched tubules and understanding their origins is an important question for the field. Here the authors develop a simple, novel assay for exploring this. By plating MCF-10A mammary epithelia cell cysts in 3D collagen, they find that these cells rapidly reorganize, break symmetry, and send out streams of cells that proliferate as they move outward, forming solid cords. This behavior is particular for collagen and is not see in Matrigel. They then proceed to use this system to explore the causal links between collagen, proliferation, directed outgrowth and integrin signaling. They find that cell proliferation coincides with outgrowth and outgrowth is blocked in proliferation is inhibited. They find that collagen is re-organized as outgrowth occurs, altering fibril organization. They then carried out a set of clever experiments to use mechanical stretch to pre-orient fibrils, and found that aggregates preferentially elongated along the fibril preferred axis. Pre-oriented fibrils also sped elongation and proliferation. Finally, blocking integrin signaling reduced/delayed elongation. I found the new system interesting, and the data intriguing. However, I had two major concerns, about the measurements presented and the causality suggested.

1. Most of the experiments involve measuring collagen fibril orientation and coherency. However, in most of the images I could not see individual fibrils (the exception, Figure 3D, was strikingly different than the rest). I would like to see a clearer description in the main text of how orientation was measured and why this is difficult to see in most images.

2. Bigger picture was the issue of causality. The authors present seemingly a linear view – integrin signaling drives proliferation and outgrowth and this triggers fibril orientation that then feeds back into the process. However, I found causality difficult to discern. While proliferation is necessary for outgrowth, its role seemed to be more likely to be permissive rather than instructive. The observation that cells would follow oriented fibrils was interesting but perhaps not surprising, and I found it difficult to tease out whether elongation triggered polarization of collagen or vice versa – I guess the authors favor a feedback process.

[Editors' note: further revisions were suggested prior to acceptance, as described below.]

Thank you for resubmitting your work entitled "Collagen polarization promotes epithelial elongation by stimulating locoregional cell proliferation." for further consideration by *eLife*. Your revised article has been reviewed by 2 reviewers, one of whom is a member of our Board of Reviewing Editors, and the evaluation has been overseen by Jonathan Cooper as the Senior Editor.

We have received comments from two of the previous reviewers, both of whom indicate the manuscript is strengthened by the addition of new experiments, text clarifications, and discussion of results in context. The reviewers request that specific representative images be added. Notably, the reviewers each agree that it is important to:

1) Provide clarification on how quantification of ERK biosensor activity was performed. Consider selecting a more representative image or using high magnification insets.

2) Provide clarification on how local regions of proliferation were calculated in 3D aggregates (i.e. did it include all cells, or only cells in contact with collagen). If needed, include a direct analysis of cells in contact with collagen in non-elongating vs elongating regions to support the conclusions.

*Reviewer #1:*

The authors have taken steps to substantially clarify the core findings from their paper, driving home their central conclusion that regional proliferation stimulated by polarized collagen directs aggregate elongation. Importantly, the authors also distinguish the conclusions of this study from prior work that apply similar methods. They have added additional experiments that address most of my concerns regarding the role of proliferation in this process and altogether the manuscript is stronger.

Where I am still not clear is the relationship between collagen and the stimulation of proliferation. This conclusion is derived from reported regional differences in cell proliferation that coincide with elongation and collagen polarization. However, this quantification could be potentially misleading, as there is also a significant difference in the number of cells with surface area contacting the surrounding collagen in non-elongating regions yet those that do appear to be proliferative (Figure 1C, 1J). Further not all instances of accumulated collagen appear to lead to elongation.

The authors clearly demonstrate (Figure 5) that polarized collagen influences the kinetics of proliferation and elongation. Perhaps this can be explained by a slightly refined model in which collagen is sufficient to stimulate cell proliferation in anlagen and its concurrent polarization/remodeling provides boundary conditions that together support the emergent behavior of elongation?

*Reviewer #2:*

The authors have addressed my main concerns in their rebuttal. The addition of new analyses strengthens the conclusion of the authors that locoregional proliferation is important for the elongation of multicellular aggregates. The data doesn't completely tease out whether this locoregional proliferation is instructive or merely permissive, but the data are more clearly interpreted and discussed in the revised manuscript.

---

## [Author Response]

Essential revisions:In this paper, the authors used reductionist models to investigate the interplay between collagen extracellular matrix and the polarized elongation of mammary epithelial anlage and provide new mechanistic insights into this process, which are potentially of broad interest. However, all three reviewers felt that there are several issues that need to be addressed before this paper can be published, most significantly those summarized below:1) All reviewers felt the data did not fully support the conclusions in several areas. Most notably, there were issues with the suggested causality of proliferation in driving analage elongation and the instructive role of polarized collagen in directing outgrowth that must be addressed.

We’ve performed additional experiments that analyse early events in anlagen elongation, especially the relationship between collagen polarization, cell proliferation and elongation. These show that:

a) When symmetry-breaking and elongation occur spontaneously, cell proliferation precedes elongation (by ~ 24 hours) and increases preferentially in those regions of aggregates that elongate.

b) Collagen polarization coincides with aggregate elongation even in these early stages (i.e. within the first 3 days of the assays).

c) Locoregional cell proliferation is stimulated further and accelerated when we induce collagen polarization extrinsically by stretching the gels.

Together, this analysis of early events reinforces the importance that enhanced loco-regional cell proliferation plays in the elongation process.

2) The authors distinguish their work by focusing on elongation rather than branching events, however the approaches and observations are similar to previous studies of branching 3D mammary explant organoids in aligned collagen matrix. How the conclusions of this work, including the observed morphogenic behavior and underlying control mechanisms, advance our conceptual understanding must be made clear.

This is a very good point and we have substantially re-written the manuscript to make the novelty of our findings clearer, especially in relationship to what has been reported before using broadly similar experimental approaches. To summarize, we believe that the contribution of our study is to identify (1) A role for loco-regional cell proliferation as a driver of epithelial anlagen elongation; and (2) A role for local reorganization of the collagen 1 matrix as a stimulator of that cell proliferation.

This contrasts with earlier studies that used similar experimental systems. For example: Brownfield et al. (Current Biology, 2013) showed that stretching collagen gels induces the elongation of embedded epithelial aggregates, but they focused on cell motility and did not test cell proliferation. Gjorevski et al. (Scientific Reports, 2013) showed that aggregates can reorganize collagen by exerting tensile forces, but focused their analysis on the impact for cell migration. Similarly, Buchmann et al. (Nat Comms, 2021) did not observe loco-regional differences in cell proliferation, but these were tested after elongation was well established, not at the early stages of symmetry-breaking as we have now done. Finally, Huebner et al. (Development, 2016) reported a role for ERK in epithelial elongation, but did not test the role of ECM organization. Thus, our findings have identified a new driver of elongation that has not been identified before.

3) Demonstrating the extent to which the findings from this work apply to mammary tubulogenesis in vivo is required.

We have expanded our Discussion to consider how our findings may apply to the more complex environment of a tissue. In particular, earlier studies have reported that the basement membrane is remodelled and thins out substantially at the tips of elongating mammary epithelial buds (and this is also seen in other models, such as the salivary gland). This would be predicted to increase cell exposure to collagen 1 in the stroma, a context that could allow the process that we have identified to operate. It will be important to test our predictions in an in vivo model, but this was not something that was feasible in the time frame of a revision. Respectfully, we suggest that it is a problem that deserves a study of its own.

Reviewer #1:Katsuno-Kambe and colleagues investigate the mechanism by which non-polarized MCF10A spheroids can be induced to extend multicellular aggregates when cultured within 3D type I collagen extracellular matrix hydrogels. Transplanting preformed aggregates into collagen is sufficient to induce asymmetric elongation of the spheroid. Branching and elongation depends on region-specific cell proliferation and multicellular outgrowths are characterized by the local accumulation of collagen matrix after the initial symmetry break. 3D hydrogel stretching and preformed aligned collagen extracellular matrix is sufficient to direct elongated outgrowth, which is dependent on proliferation, β1 integrin and ERK signaling. While the work is quantitative and the imaging is of high quality, chemical and physical factors that affect the long axis bias of the developing mammary epithelium have been intensively studied in vivo and in vitro, and it is not clear that this work offers a significant conceptual advance. The study also would need validation of observations from their engineered in vitro systems with relevant aspects of in vivo conditions.Methods and conclusions from this work are similar to those previously reported by the Bissell group (Brownfield et al. Current Biology, 2013). In this study, unpolarized primary murine mammary explants were embedded within 3D collagen hydrogels and axially stretched using a deformable PDMS-based device. Collagen stimulated explant branching and outgrowth, which required collagen fiber polarization. The authors do reference this study, but argue its focus is on branching rather than elongation. However, given the nearly identical methods, analyses, and morphogenic conclusions, it is unclear how this present work is distinct. Similarly, it is unclear if the mechanisms derived from the reductionist system here are relevant to mammary development/tubulogenesis in vivo.1) Recent and previous work from the Bissell and Nelson groups have explored the role of collagen fiber polarization and/or axial alignment during patterned elongation of the developing mammary gland. While the role collagen still remains poorly understood, it will be important to reconcile these studies with the work presented here, namely by demonstrating the relevance of the findings from this work to mammary tubulogenesis in vivo.

i) Although the Bissell and Nelson groups have used broadly similar experimental systems, they did not identify the important role for cell proliferation, as we have done. In particular, it was striking that proliferation in our experiments was increased *preferentially* in those regions of the epithelial aggregates that underwent elongation; and this appears to be stimulated by the local re-organization of the collagen 1 matrix. Such loco-regional stimulation of cell proliferation was not reported in earlier studies. For example, Brownfield et al. (Current Biology, 2013; Bissell group) showed that stretching collagen gels induces the elongation of embedded epithelial aggregates, but they focused on how this might occur through cell motility and did not test cell proliferation. As well, Gjorevski et al. (Scientific Reports, 2013; Nelson group) showed that aggregates can reorganize collagen by exerting tensile forces, but also focused their analysis on its impact on cell migration.

Thus, we think that the notion that collagen 1 reorganization can promote elongation by stimulating cell proliferation is a key message of our study that opens new avenues for the field. However, we did not present this with adequate focus in the original manuscript. So, we have thoroughly rewritten the revised manuscript to make this more explicit. We thank all the reviewers for prompting us to do a better job.

ii) We have expanded the Discussion to consider how our findings may be relevant for the more complex environment of tubulogenesis in vivo. Of course, our system is explicitly reductionist and does not capture the complexity of the ECM in vivo, especially in terms of other ECM components as well as a basement membrane. However, it is noteworthy that studies in vivo have reported important roles for collagen 1 during mammary tubulogenesis (e.g. Kelly et al., Differentiation 1995) and that the basement membrane often thins in regions of elongation e.g. Williams and Daniel, Dev Biol 97, 274, 1983; Silberstein and Daniel, Dev Biol 90, 215, 1982), making these more likely to engage with stromal collagen 1. We have included this in the Discussion as ways in which our findings may have impact in vivo. While it would be important to test these concepts in the more complex environment of a tissue, we respectfully felt that this was beyond the feasible scope of a revision.

2) Methods and conclusions from this work are similar to those reported by the Bissell group (Brownfield et al. Current Biology, 2013). In this study, unpolarized primary murine mammary explants were embedded within 3D collagen hydrogels and axially stretched using a deformable PDMS-based device. Collagen stimulated explant branching and outgrowth, which required collagen fiber polarization. The authors do reference this study but argue its focus is on branching rather than elongation. However, given the nearly identical methods, analyses, and morphogenic conclusions, it is unclear how this present work is distinct.

This is a very good point, one that indicates that we had not adequately focused the message of our manuscript. As noted above, although the experimental models may be similar, what is distinct in our findings is the role for loco-regional cell proliferation as a driver of epithelial elongation. In contrast, both Brownfield et al. (2013) and Gjorevski et al. (2013) focused on cell migration as the basis for how reorganizing collagen might affect the morphogenetic process of epithelial elongation. Recently, Buchmann et al. (Nat Comms, 2021) also focused on migration; they reported that proliferation occurred throughout their aggregates, but did not examine early stages of symmetry-breaking as we have done. So, we believe that our findings introduce a new driver for elongation. And we have fully rewritten the manuscript to make this clearer and more explicit.

3) "In contrast, Rac1 inhibitor treatment did not affect aggregate elongation and collagen fibril rearrangement (data not shown)". These data should be shown, as it conflicts with the conclusions of the Brownfield study. It should be noted that the Rac1 inhibitor NSC23766 only blocks a subset of Rac1 interactions with Rac1GEFs.

We have now included this data (Figure 2-Supplemental Figure 2). It should be noted that Brownfield et al. tested the potential role for Rac 1 by overexpressing a constitutively-active transgene. It is possible that the differences in our conclusions derive from our use of an inhibitor (albeit one that does not target Rac1 directly). Our data are also consistent with the findings of Gjorevski et al. (2013) and Buchmann et al. (Nat Comms, 2021), who reported that inhibiting ROCK disrupted the ability of cells to reorganize their surrounding collagen matrix.

4) From Figure 2A, is the gel coherency metric capturing fibril alignment, or the localized accumulation of collagen fluorescent intensity? Is collagen accumulation a function of cells pushing/expanding out into the matrix, or from active fiber recruitment to the outgrowth surface?What determines whether localized collagen intensity leads to branching and outgrowth? In Figure 2A, there are two distinct regions of polarized collagen, but only one contains outgrowth. There is an overall variability in the brightness, contrast, and resolution of the representative collagen images, making it difficult to interpret across experiments. For instance, polarized collagen appears to be present in the blebbistatin treated outgrowths, however it is difficult to discern due to the different magnification and non-uniform hydrogel intensity.

i) We measured collagen coherency using Orientation J, which calculates the local orientation and isotropy for each pixel in an image based on the structure tensor for that pixel. The structure tensor is evaluated for each pixel of the given image by calculating the spatial partial derivatives by using (a cubic B-spline) interpolation. The local orientation and isotropy for each pixel are computed based on the eigenvalues and eigenvectors of the structure tensor.

We therefore used coherency as a measure of fibril co-alignment, which we interpret to reflect bundling, rather than purely collagen accumulation. It should be noted that we used this approach because our experiments required low-magnification, long working distance lenses to capture aggregates and their surrounding matrix. Therefore, we could not resolve individual collagen fibrils (except for the floating, cell-free gel in Figure 3D, where we could image close to the surface).

ii) At this point, our data don’t allow us to determine the cellular mechanism(s) that are responsible for the collagen reorganization. Given the slow time course of the phenomena, it is entirely possible that multiple mechanisms (including tugging forces and pushing forces) may contribute. With respect, we feel that it is somewhat peripheral to the key point of our study, which is how interplay between cell proliferation and collagen reorganization drives epithelial elongation. It will be important to elucidate this issue, but may be better suited to an independent study.

iii) Unfortunately, we can’t address the question of what may control the choice of branching vs elongation, because our analyses focused purely on the elongation process. In our experience, collagen 1 gels commonly show some variability in density (as reflected in differences in the intensity of labelling), even in the absence of cells. This prompted us to compare organization features of the ECM in regions of interest that were placed either adjacent to the parts of aggregates which showed elongation or those parts that did not.

5) Is proliferation truly driving this behavior? In the representative images (Figure 1C, 1H) it appears that all peripheral MCF10A making basal matrix contact are Ki67 positive. It is a bit difficult to tell when proliferation arresting inhibitors were added in the evolution of these processes. Does acute addition of mitomycin halt, or regress, established outgrowths?

i) In our original experiments, we added mitomycin C or aphidicolin when aggregates were first transplanted into collagen 1 gels, so proliferation inhibitors were present throughout the experiments. To test whether proliferation might be required to sustain elongation once it had begun, for this revision we added the drugs at day 3, when elongation had begun. Even though the aggregates had already broken symmetry, inhibition of proliferation significantly reduced further elongation, consistent with the idea that proliferation is necessary to sustain elongation. In the revised manuscript, we:

a) Clarify the timing for when proliferation inhibitors were added.

b) Include this new data.

ii) To summarize the evidence for a important role of stimulated cell proliferation, we find that:

a) There is a regional difference in cell proliferation that coincides with elongation. Greater proliferation was found in the regions of aggregates that elongate compared with those regions that do not elongate. We measured this by quantitating both Ki-67 staining and also counting cell divisions by live-cell imaging.

b) Stimulation of elongation by stretching gels was also associated with stimulation of cell proliferation in a loco-regional pattern that paralleled the elongation.

c) Both spontaneous and stretch-stimulated elongation were reduced by blocking cell proliferation. Together, we feel that these data argue for a necessary role of proliferation in driving elongation in our experiments. We don’t exclude the possibility that other processes, including polarized cell divisions, contributed to the elongation process. But they clearly did not compensate to sustain elongation when proliferation was blocked.

ii) The relationship between proliferation and cell migration is something that we have analysed in greater detail for this revision (prompted by comments from Reviewer 2). Of note, migratory cells were found throughout elongating aggregates, both in the regions that were elongating and in those that did not elongate. In control cultures, we found no regional difference in migration speed but in the elongating regions cells tended to be a little more persistent in their movement and oriented with the principal axis of the aggregate, which is perhaps to be expected. Nor was there any change in the regional patterns of cell speed when we inhibited proliferation with Mitomycin C or aphidicolin. Surprisingly, these proliferation inhibitors slightly decreased the persistence of movement and decreased the alignment with the principal axis of the aggregates. This suggested that features of coordinated collective migration were compromised when proliferation was inhibited. Therefore, by some means coordination of migration may also mediate the impact of regional cell proliferation on epithelial elongation.

6) Similarly, do stretch-induced elongations regress after removal from stretch and transfer to floating culture?

We tried to do this experiment, but it was technically unsuccessful. This is because the gels shrank when they were transferred into floating culture, making it difficult to visualize the aggregates adequately with our microscope set up.

7) AIIB2 implicates β1 integrin, but not α2β1. The authors should check to see if fibronectin deposition is stimulated by stretch, and is fibronectin present at stochastic outgrowths in non-patterned collagen?

i) Fibronectin: We found that fibronectin stained throughout the interfaces between aggregates and the collagen matrix. This appeared to be uniform in control aggregates that elongated spontaneously, without any evident regional variation in regions that elongated compared with regions that did not elongate (Figure 2—figure supplement 1B). Nor did we see any evident increase in fibronectin staining when cultures were stretched (Figure 3—figure supplement 2).

ii) A good point about AIIB2 implicating β1 integrin generally rather than α2β1 specifically: we’ve corrected our text accordingly.

8) "This regional patterning would be necessary if proliferation drives elongation, as a global increase in cell number would cause aggregates to expand isotropically." It is important to acknowledge that a critical regulatory feature missing from this model is a developed basement membrane. in vivo, the local degradation and remodeling of basement membrane mechanically permit region specific expansion of developing branched tissues, such as in the embryonic salivary gland (Harunaga et al. Dev. Dyn., 2014).

This is a good point and one that also speaks to how findings from this reductionist system might be translated into the complex environment of the native tissue (as discussed in our Response ii to point 1 above). Similarly, local thinning and loss of the basement membrane has been reported at the tips of elongating mammary tubules (e.g. Silberstein and Daniel, Dev Biol, 90, 215, 1982; Williams and Daniel, Dev Biol 97, 274, 1983), suggesting something similar may occur, which locally reduces the impact of the basement membrane, exposing the cellular compartment to collagen in the surrounding matrix.

9) "The decision to branch is recognized to be a critical checkpoint that is controlled by developmental signaling." It is unclear what distinction the authors are trying to make. There is robust literature suggesting cell nonautonomous mechanisms regulating branching events, notably localized extracellular matrix deposition and heterotypic cell-cell interactions.

Here we were trying to make the point that less is known about what regulates elongation, compared to what is known about branching. We’ve modified the text to make this clearer; it now reads:

“Tubulogenesis is a highly regulated phenomenon. The decision to branch is recognized to be a critical checkpoint that is controlled by developmental signals and cell-cell and cell-ECM interactions. The elongation of tubule precursors is also thought to be a regulated process that is controlled by receptor tyrosine kinases and other signaling pathways (Costantini and Kopan, 2010; Gjorevski and Nelson, 2010; Sternlicht et al., 2006).”

10) The aspect ratios of the images in Figure 5D appear to be off.

This reflects the different magnifications that were used to capture the full size of the aggregates (the one on top was much longer than that at the bottom). The scale bars, as printed look the same, but are different, and we omitted to define the scale bars in the legend. That was a source of confusion that we’ve corrected.

Reviewer #2:The formation of mammary gland tubules begins as non-polarized cellular aggregates, called anlage, which elongate as solid multicellular cords that eventually form lumens. Katsuno-Kambe et al. investigate the role of the extracellular matrix during the process of elongation, making use of 3D culture models of mammary epithelial MCF10A cells. These cells have previously been shown (e.g Krause et al., Tissue Eng Part C Methods 2008, and others) to form multicellular aggregates that polarize and eventually form lumens when cultured in Matrigel, and when embedded in Collagen I (a major component of the stromal environment) form elongated, solid cords of non-polarized cells. Therefore, to capture the process by which aggregates break symmetry and become elongated, the authors first seeded MCF10A cells in Matrigel until proliferation arrested (resembling anlage), after which aggregates where isolated and embedded into collagen I gels. The authors show increased proliferation of cells in elongating areas of aggregates, and that these elongated aggregates are not formed when proliferation is inhibited. The authors further show that this elongation is accompanied by collagen reorganization into polarized bundles around elongating areas, and by artificially polarizing collagen by application of external stretch show collagen polarization can increase proliferation and guide the direction of elongation. Finally, the authors demonstrate aggregate elongation requires beta1-integrin adhesion and downstream ERK activity.The strength of this manuscript is the use of a reductionist approach to investigate the role of collagen alignment in the elongation of epithelial anlage, by transplanting MCF10A cell clusters from matrigel into Collagen I gels, and artificially manipulating collagen polarization by stretching these gels. In addition, the exploration of the underlying mechanism of elongation, requiring proliferation and integrin-ERK signaling, is of potential great interest to understanding the role of cues derived from the extracellular matrix in epithelial morphogenesis.The main weakness of this study is that the data still insufficiently supports several conclusions, in particular on proliferation being a key driver of anlage elongation, and on the reciprocal interaction between proliferation and collagen polarization during this process. The induction of proliferation is only analyzed 8 days after cell clusters have been transferred to Collagen gels, and thus after symmetry break and elongation have already occured. Analyses of proliferation and collagen polarization during the different stages of symmetry break and elongation, and detailed analyses of the distribution of proliferative cells during these stages, are lacking. This is in particular relevant as also in non-elongated areas, cells that are in contact with Collagen appear to be proliferative. Alternative driver mechanisms, for instance the induction of collective migration, are not fully tested and it remains unclear whether proliferation is merely required for, or truly a driver event of aggregate elongation.The authors investigate a role for integrin signaling and downstream ERK in the induction of proliferation and aggregate elongation in response to collagen polarization. While an integrin β 1 inhibitory antibody and ERK inhibitor indeed prevents elongation, key unanswered questions are whether this pathway is indeed activated by polarized collagen during elongation (i.e. in selective regions of the cell clusters) and whether it is required for the proliferation response itself.Despite the elegance of the reductionist approach in MCF10A cells in vitro, it remains unclear to what extent these findings generally apply to mammary tubulogenesis under physiological conditions in vivo, which remains largely undiscussed in the manuscript.Comments for the authors:1. The authors conclude that cell proliferation is a key driver of epithelial anlage elongation. However, the induction of proliferation is only analyzed 8 days after transfer of cell clusters to Collagen I, after symmetry break and elongation have taken place. And here, analysis of proliferation is limited to cells in the elongated areas versus the remaining cell cluster. It is essential to include analyses of proliferation during the different stages of elongation (i.e. before and upon initial symmetry break and at different time points of elongation), as well as detailed analyses of the distribution of proliferative cells during these stages. This is in particular relevant as from the images (e.g. Figure 1C) it is apparent that also in the non-elongated areas the cells on the outside of the cell cluster (thus in contact with Collagen I) are proliferating (e.g. Figure 1C), yet do not form elongated strands (which also does not support the authors conclusions that "proliferation is selectively increased in regions of aggregates that were elongating"). Finally, analysis of the induction of proliferation should also include a control in which cell clusters are transferred back to matrigel.

i) We have performed additional experiments to analyse proliferation early in the elongation process.

a) In order to assess when overall changes in proliferation became evident, we measured the proportion of cells that were Ki67-positive in the first 3 days of culture. An increase in Ki67-positivity was first evident on day 2 which was, interestingly, about a day before elongation was first detected.

b) To examine if there were locoregional differences in proliferation rates in these early aggregates, at day 3 we examined aggregates that had broken symmetry and begun to elongate (which we defined experimentally as a symmetry ratio of >1.5). We compared the proportion of Ki67-positive cells in the elongating regions of aggregates with the proportion found in the non-elongating parts of the aggregates. This showed a significant increase in the proportion of Ki67-positive cells in the elongating regions compared with the non-elongating regions. These new data lead us to conclude that loco-regional differences in cell proliferation are evident early in the elongation process, as well as later (as we had initially demonstrated).

ii) Furthermore, as the reviewer notes, cell proliferation was certainly seen in regions of aggregates that did not elongate. However, proliferation was greater in the elongating areas. So, it is the loco-regional difference in proliferation that seemed to be significant in our experiments. We have endeavoured to clarify this point in the revised manuscript.

iii) We also performed the control experiment where aggregates were extracted from Matrigel, then re-embedded in Matrigel. The data are shown in Figure 1—figure supplement 1C: we found that the aggregates did not elongate and, although Ki67 positive cells were evident, they appeared to be distributed randomly in the aggregates.

2. Alternative driver mechanisms, in particular cell migration, are limited to analysis of migration speed that is similar in different regions of the aggregate. This does not exclude that induction of directional, collective migration could be the main driving event underlying elongation, which remains unexplored. As such, it remains undetermined whether proliferation is merely required for, or truly a driver event of aggregate elongation.

We thank the reviewer for this suggestion and took the opportunity to measure a number of other parameters of cell motility for this revision (Figure 1—figure supplement 2B-H). In addition to cell speed, we also measured the straightness of the tracks (as an index of persistence) and the orientation of tracks relative to the principal axis of the aggregates. For these parameters we compared cells in the elongating regions of aggregates with cells in the non-elongating regions. There was no difference in migration speed between these two areas, but cells in the elongating areas migrated slightly more persistently and tended to orient themselves with the axis of elongation. Such an enhanced orientation of the migrating cells would be predicted to facilitate elongation.

Surprisingly, we found that, although it did not affect the speed of movement, inhibition of proliferation decreased the apparent persistence of movement and also caused cells to orient with less fidelity to the principal axis of the aggregates. Such changes in orderly migration could also have contributed to the defect in elongation. However, the implication is that these features of migration depend in some way upon cell proliferation. Our current data do not allow us to explain why this occurs. One possibility is that reorganization of the collagen matrix, which we show to be influenced by cell proliferation, influences cell movement. Polarized collagen could be regarded as a 3D micropattern that surrounds the elongating regions of the aggregates. Irrespective of the mechanism responsible, we suggest that this new data reinforces the concept that regional stimulation of proliferation is key to epithelial elongation in our system: and it may work both directly (by increasing cell number preferentially in the elongating areas) and also (directly or indirectly) by enhancing features of orderly cell migration that would also promote elongation.

3. The authors mention in the discussion that "the initial step of symmetry-breaking was associated with reorganization of collagen fibrils along the aggregate (…) which became increasingly bundled and aligned as elongation proceeded". However, data showing these changes in collagen during the different steps are not presented in the manuscript. This is in particular important as the authors conclude that collagen polarization induces proliferation, and therefore the authors should include a more detailed analyses of both processes (collagen reorganization and proliferation) over time after cell clusters are transferred to Collagen gels, and how they are associated with each other in individual cell clusters.In addition, can the authors explain the induction of proliferation in cells in contact with collagen in the non-elongating areas (e.g. Figure 1C, see comment 1), as this suggest contact with collagen is sufficient to increase proliferation and collagen polarization is not required?

i) We have now examined collagen organization earlier in the elongation process (Figure 2E). For this, we analysed aggregates within the first 3 days of the assays. Because aggregates do vary in the time when they broke symmetry we subdivided them based on their symmetry ratio, comparing the coherency of the collagen gel around aggregates that remained rounded (symmetry ratio ~ 1) or around those with progressively greater symmetry ratios. We found that gel coherency increased (which we interpret as reflecting collagen bundling) with the increase in symmetry ratio. This confirmed that collagen reorganizes during the earliest stages of aggregate elongation.

ii) It is important to emphasize that cell proliferation was observed throughout the aggregates, both in regions that failed to elongate as well as those that did elongate. This implies that it is the regional difference is one of degree, not all or nothing. In other words, the key is what may increase proliferation in the areas of aggregates which elongate. Therefore, it becomes noteworthy that the regions that show increased proliferation are those which are adjacent to the more polarized collagen. This was evident in aggregates that broke symmetry spontaneously. Also, when we induced collagen polarization by stretching the gels, proliferation was stimulated preferentially in the elongating areas.

Thus, we conclude that the regional increase in cell proliferation associated with elongation is not being stimulated by contact with collagen *per se*, but rather by some additional property of the collagen gels that is associated with their reorganization. What that property may be is not something which we can answer at the present time; and we’d respectfully suggest that it may be more suitable to a separate, future project. Instead, we discuss some possibilities – including changes in stiffness of the collagen matrix – in greater depth and explicitness in the revised discussion.

4. In line with the above comment, the conclusion that collagen polarization promotes proliferation is largely based on the experiments in which collagen gels were stretched. However, the effect on proliferation is again only assessed following 8 days of culture in these gels, and thus after aggregate elongation. To support their conclusion, proliferation should be analyzed at early time points following stretching. This would also address whether the earlier symmetry break observed after stretching is a consequence of increased/faster induction of proliferation, or involves the ability of already proliferating cells to drive epithelial elongation (which will affect the interpretation of these data).

This was a good point and we have now done these experiments. We find that enhanced proliferation is detectable even one day after stretching. As shown in Figure 5A, proliferation begins earlier and to a greater degree in stretch-stimulated aggregates. This reinforces our interpretation that stimulation of proliferation is a driver of elongation, and we thank the reviewer for this suggestion.

5. The authors demonstrate that inhibition of proliferation prevents aggregate elongation. It is difficult to interpret from these experiments whether proliferation could truly be a key driver of elongation, as aggregates appear to remain much smaller following mitomycin C and Aphidicolin treatment (Figure 1H). It should be shown whether in the absence of the inhibitors, symmetry break and elongation is induced at this aggregate size (and thus whether the observed effect represents a direct or indirect of the inhibitors on elongation). In addition, proliferation inhibitors should also be given after the symmetry break has taken place, to test whether this prevents protrusions that are already present from elongating further.

a. We have observed small control aggregates (with sizes comparable to those of proliferation-inhibited aggregates) that break symmetry. However, aggregates tended to break symmetry at different times in culture (and therefore at different sizes). So, we have not endeavoured to quantitatively compare the sizes when symmetry broke in controls vs proliferation-inhibited cultures. Accordingly, we include representative images (Author response image 1) , but would be happy to add them to the manuscript if the reviewer and editor consider it to be warranted.

**Author response image 1. sa2fig1:** 

b. To test the impact of inhibiting proliferation after symmetry had broken, we added drugs at day 3, when symmetry breaking had occurred (Figure 1M). Elongation was significantly inhibited, implying that cell proliferation contributes to sustaining aggregate elongation once symmetry has broken. This observation reinforces our interpretation that proliferation plays a critical role in the elongation process, rather than differences in aggregate size.

6. The authors investigate a role for integrin signaling and downstream ERK in the induction of proliferation and aggregate elongation in response to collagen polarization. While indeed both integrin beta1 and ERK activity are necessary for aggregate elongation, whether this pathway is directly involved in aggregate elongation remains unclear, and analysis of cell proliferation is lacking. The authors should test with available antibodies whether this pathway is specifically induced by polarized collagen (by analyzing different regions of the aggregate during symmetry break and elongation). In addition, effect of inhibitors on the level and distribution of Ki67+ cells should be shown.

We performed a number of additional experiments in response to these suggestions.

i) To determine if ERK was, indeed, a downstream element in an integrin-dependent pathway, we used the ERK location sensor to measure how ERK signaling was affected by the β1-integrin-blocking antibody, AIIB2, when elongation was stimulated by stretching of the gels. We found that the proportion of cells showing ERK activation was significantly reduced by AIIB2, both early (day 2) and late (day 7) in the assays (Figure 7H, I). Thus, we consider that β1-integrin is driving proliferation via ERK.

ii) To test if there were loco-regional differences in ERK activation, we measured the proportion of ERK-activated cells in the elongating and non-elongating areas of aggregates (Figure 6A-C). Consistent with what we had observed for proliferation, a greater proportion of cells showed ERK activation in the elongating areas, than in the non-elongating areas. This loco-regional difference was evident when we measured ERK activation in cells throughout the aggregates (Figure 6B) and when we only analysed those at the surface, which were in contact with the collagen (Figure 6C).

iii) We measured the effect on proliferation of inhibiting ERK with FR180204 (Figure 6D,E) or blocking β1-integrin with AIIB2 (Figure 7D). These new data show that proliferation was reduced by these inhibitors.

7. To control that effects observed in stretched collagen gels are truly mediated by the induced collagen polarization, the authors allowed gels to float in the medium to revert gel isotropy to that of unstretched gels. While collagen polarization is indeed reverted, cells cultured in floating gels will not able be able to generate stresses themselves (e.g. Halliday et al., 1995). From these experiments it can therefore not be concluded whether it is truly the collagen polarization or stresses generated by the cells that are needed for aggregate elongation.

This is a very good point, and one that indicates something which warrants clarification. We think that collagen polarization and stress-generation by cells are closely linked processes. When aggregates spontaneously break symmetry, we’d hypothesize that anisotropies in contractile tension are responsible for initiating collagen polarization. Conversely, the stretched gels that we use to exogenously polarize the gels are likely to apply stresses to the cell. Furthermore, as implied by the reviewer it is possible that cells generate different stresses when they are exposed to polarized compared with unpolarized collagen, and this “internal” stress is what is promoting proliferation.

However, the key point of our paper is that collagen polarization promotes elongation by stimulating cell proliferation in the areas of aggregates that are adjacent to the polarized collagen. The source of the stress that polarizes the collagen is less important, in this perspective, than the fact that polarization promotes proliferation. This was not clear in the original manuscript and we’ve endeavoured to correct this flaw in the revision.

8. It remains unclear to what extent the findings from this manuscript apply to mammary tubulogenesis in vivo, which remains undiscussed in the manuscript. While some experiments may be difficult to verify in vivo (e.g. to monitor collagen alignment), it would be relevant to at least to know whether in vivo mammary epithelial cells indeed come in contact with collagen I during anlage elongation, and preferably also show whether this is accompanied by a specific proliferation response.

Here we can draw guidance from the literature, which has reported that: (a) collagen is a major component of the ECM in morphogenetically-active mammary buds (Keely et al., Differentiation, 59: 1, 1995) and, also, (b) the basement membrane thins substantially at those growing tips, which would facilitate contact between cells and collagen, as reduce the mechanical impact of the basement membrane, compared with the surrounding ECM (Silberstein and Daniel, Dev Biol, 90, 215, 1982; Williams and Daniel, Dev Biol 97, 274, 1983). We’ve included discussion of this point in the final paragraphs of the Discussion, to highlight the physiological implications of our in vitro findings. With due respect, we feel that to extend our current observations into an in vivo model would be beyond the reasonable scope of a revision and better suited to a future study.

Reviewer #3:Understanding tissue morphogenesis requires an understanding of underlying molecular and mechanical mechanisms. Many tissues form a network of branched tubules and understanding their origins is an important question for the field. Here the authors develop a simple, novel assay for exploring this. By plating MCF-10A mammary epithelia cell cysts in 3D collagen, they find that these cells rapidly reorganize, break symmetry, and send out streams of cells that proliferate as they move outward, forming solid cords. This behavior is particular for collagen and is not see in Matrigel. They then proceed to use this system to explore the causal links between collagen, proliferation, directed outgrowth and integrin signaling. They find that cell proliferation coincides with outgrowth and outgrowth is blocked in proliferation is inhibited. They find that collagen is re-organized as outgrowth occurs, altering fibril organization. They then carried out a set of clever experiments to use mechanical stretch to pre-orient fibrils, and found that aggregates preferentially elongated along the fibril preferred axis. Pre-oriented fibrils also sped elongation and proliferation. Finally, blocking integrin signaling reduced/delayed elongation. I found the new system interesting, and the data intriguing. However, I had two major concerns, about the measurements presented and the causality suggested.1. Most of the experiments involve measuring collagen fibril orientation and coherency. However, in most of the images I could not see individual fibrils (the exception, Figure 3D, was strikingly different than the rest). I would like to see a clearer description in the main text of how orientation was measured and why this is difficult to see in most images.

We appreciate this point of clarification. Individual collagen fibrils were difficult to image in our experiments because low magnification imaging was necessary, for two reasons: (a) We were endeavouring to capture whole aggregates and their surrounding matrix in 3D videos; and (b) higher magnification lenses did not provide the depth of focus needed to image aggregates deep within the gels. (On this point, the fibrils are evident in Figure 3D because this floating gel allowed us to access the gel surface. In contrast, when gels were embedded – including after they had been stretched – then we had to image deeper.)

Accordingly, we analysed collagen organization using Orientation J, which calculates the local orientation and isotropy for each pixel in an image based on the structure tensor for that pixel. The structure tensor is evaluated for each pixel of the given image by calculating the spatial partial derivatives by using (a cubic B-spline) interpolation. The local orientation and isotropy for each pixel are computed based on the eigenvalues and eigenvectors of the structure tensor.

2. Bigger picture was the issue of causality. The authors present seemingly a linear view – integrin signaling drives proliferation and outgrowth and this triggers fibril orientation that then feeds back into the process. However, I found causality difficult to discern. While proliferation is necessary for outgrowth, its role seemed to be more likely to be permissive rather than instructive. The observation that cells would follow oriented fibrils was interesting but perhaps not surprising, and I found it difficult to tease out whether elongation triggered polarization of collagen or vice versa – I guess the authors favor a feedback process.

In revising the manuscript, we have taken the opportunity to perform additional experiments and also to present the data and our interpretation with greater clarity. In particular, we argue that polarized collagen exerts its impact on elongation by promoting proliferation in the parts of the cell aggregate that are adjacent to the polarized collagen. Our new data show that:

a) When aggregates elongate spontaneously, cell proliferation is detectable ~ 24 hours before elongation. And even in these early stages proliferation is greater in the elongating parts of aggregates, rather than in the non-elongating parts of the aggregates. Since inhibitors of proliferation reduce elongation, these correlations suggest that selective increase of proliferation within elongating areas is necessary for that elongation to occur.

b) Cell proliferation is also stimulated when we induce collagen polarization extrinsically, by stretching the gels (Figure 5). Strikingly, we find that an increase in cell proliferation occurs earlier with stretch-polarization than when aggregates are allowed to break symmetry and polarize spontaneously. And, as seen with spontaneous elongation, proliferation is greater in the elongating areas of the aggregates. We take this as evidence that polarization of collagen can stimulate cell proliferation in a loco-regional fashion.

c) We have performed further analyses to characterize cell migration within aggregates in greater detail. Interestingly, we find that although inhibitors of mitosis did not affect the speed of cell migration, they disrupted the directionality of migration. Cells treated with aphidicolin or Mitomycin C did not orient with the principal axis of the aggregates as well did control cells. This suggests that an effect on directional migration may have contributed to the disruption of epithelial elongation that occurs when proliferation is inhibited.

We propose that these data identify the capacity for polarized collagen to stimulate cell proliferation in the parts of the aggregates proximate to the polarized gel. And this is necessary for elongation to occur effectively. We have endeavoured to make this point more clearly in this revision than we did earlier, because we consider this to be a new insight into how elongation occurs. Earlier studies, using experimentally-similar systems, principally focused on how matrix organization might affect cell locomotion to promote elongation. Migration is unarguably important, but our data suggest that it is not enough. Migration did not compensate when proliferation is inhibited and, indeed, may be supported by proliferation in some way.

Finally, we agree that feedback is likely to be important when aggregates break symmetry and extend spontaneously. We found that collagen polarization was reduced when cell proliferation was blocked, suggesting that polarization may be initially induced by some mechanical change associated with proliferation. We discuss this briefly in the revision. Possibly, local variations in cell proliferation induce anisotropies of force to being to polarize the collagen. However, this is largely speculative at the present time: more detailed temporal and spatial analysis that compares proliferation with elongation and collagen polarization will be needed to answer this question. We respectfully suggest that these experiments would make the subject of a separate study in themselves.

[Editors' note: further revisions were suggested prior to acceptance, as described below.]

We have received comments from two of the previous reviewers, both of whom indicate the manuscript is strengthened by the addition of new experiments, text clarifications, and discussion of results in context. The reviewers request that specific representative images be added. Notably, the reviewers each agree that it is important to:1) Provide clarification on how quantification of ERK biosensor activity was performed. Consider selecting a more representative image or using high magnification insets.

In our quantitation, we used nuclear localization of the ERK biosensor as our measure of ERK activation. This was not made clear in the text and we have now made it more explicit in this revision.

And to assist the reader, we have included higher magnification views of Figure 6a, to identify cells that we have scored as “active” (i.e. with nuclear exclusion) as well as those that we scored as “inactive” (i.e. where there was both nuclear and cytoplasmic staining). This scoring criterion is also explicitly mentioned in the Results and Figure caption.

2) Provide clarification on how local regions of proliferation were calculated in 3D aggregates (i.e. did it include all cells, or only cells in contact with collagen). If needed, include a direct analysis of cells in contact with collagen in non-elongating vs elongating regions to support the conclusions.

For proliferation we measured the proportion of cells which were Ki-67 positive throughout the regions of interest. In this revision we further analysed only the cells that were in direct contact with the collagen. This confirmed the result that we obtained when all cells were analysed – i.e. the proportion of proliferating cells was greater in the elongating regions than in the non-elongating regions. This data is now included in Figure 1—figure supplement 2A, corresponding to Figure 1C – which is where we establish the basic observation that proliferation differs in elongating vs non-elongating regions of aggregates.

Reviewer #1:The authors have taken steps to substantially clarify the core findings from their paper, driving home their central conclusion that regional proliferation stimulated by polarized collagen directs aggregate elongation. Importantly, the authors also distinguish the conclusions of this study from prior work that apply similar methods. They have added additional experiments that address most of my concerns regarding the role of proliferation in this process and altogether the manuscript is stronger.Where I am still not clear is the relationship between collagen and the stimulation of proliferation. This conclusion is derived from reported regional differences in cell proliferation that coincide with elongation and collagen polarization. However, this quantification could be potentially misleading, as there is also a significant difference in the number of cells with surface area contacting the surrounding collagen in non-elongating regions yet those that do appear to be proliferative (Figure 1C, 1J). Further not all instances of accumulated collagen appear to lead to elongation.

To pursue this, we further analysed the data set of Figure 1C and J to measure proliferation just in the cells that were in contact with the collagen. We found that the proportion of cells which were Ki-67, was greater in regions of aggregates that were elongating, compared to the non-elongating regions. This confirmed our earlier analysis that encompassed all the cells in the respective regions. It further reinforces our conclusion that the rate of cell proliferation was greater in the elongating regions of aggregates.

This data (which extends the analysis in Figure 1C and 1J) is now included in Figure 1—figure supplement 2A.

The authors clearly demonstrate (Figure 5) that polarized collagen influences the kinetics of proliferation and elongation. Perhaps this can be explained by a slightly refined model in which collagen is sufficient to stimulate cell proliferation in anlagen and its concurrent polarization/remodeling provides boundary conditions that together support the emergent behavior of elongation?

This is a very good idea, which we have happily stolen for the final paragraph of our discussion (lines 542-544):

“The consequent loco-regional collagen polarization then stimulates further cell proliferation in the adjacent parts of the aggregate to sustain elongation. Biochemical contact with collagen may provide a basal stimulus for cell proliferation, which is further enhanced where collagen becomes polarized to provide a boundary condition that promotes elongation.”